# Causal Discovery and Inference through Next-Token Prediction

**Eivinas Butkus**[1,2]
eivinas.butkus@columbia.edu

**Nikolaus Kriegeskorte**[1,2]
n.kriegeskorte@columbia.edu

[1]Columbia University
[2]NSF AI Institute for Artificial and Natural Intelligence

## Abstract

Deep neural networks have been criticized as fundamentally *statistical* systems that fail to capture causal structure and perform causal reasoning. Here we demonstrate that a GPT-style transformer trained for next-token prediction can simultaneously discover instances of linear Gaussian structural causal models (SCMs) and learn to answer counterfactual queries about those SCMs. First, we show that the network generalizes to counterfactual queries about SCMs for which it has seen interventional data but not any examples of counterfactual inference. The network must, thus, have successfully composed discovered causal structures with a learned counterfactual inference algorithm. Second, we decode the implicit "mental" SCM from the network's residual stream activations and manipulate it using gradient descent with predictable effects on the network's output. Our results suggest that statistical prediction may be sufficient to drive the emergence of internal causal models and causal inference capacities in deep neural networks.

## 1 Introduction

How can an AI system discover and learn to reason about causes and effects—the mechanisms that remain invariant under local interventions [43, 35]? Pearl [36, 38] has argued that deep neural networks (DNNs) trained using prediction objectives are intrinsically limited in their causal reasoning capacities. His argument rests on Pearl's Causal Hierarchy (PCH) [2], also known as the "Ladder of Causation" [38]. PCH describes three levels of causal capabilities—associational ($\mathcal{L}_1$), interventional ($\mathcal{L}_2$), and counterfactual ($\mathcal{L}_3$)—and implies that answers to higher level queries are generally underdetermined by data or information from lower levels. According to Pearl [36], this hierarchy implies that DNNs trained in a "statistical mode" to predict passive observations can only master associations ($\mathcal{L}_1$) and are prevented from reasoning about actions, experiments, and explanations ($\mathcal{L}_2$ and $\mathcal{L}_3$).

If true, Pearl's claim would have important theoretical and practical implications regarding the causal abilities of large language models (LLMs) and other foundation models. Such models are typically pretrained using statistical objectives to predict held-out portions of passive streams of data [8, 40, 39, 6, 5]. For example, LLMs are trained to predict the next token in snippets drawn from a diverse corpus of text. Following Pearl's reasoning [36], at least initial pretrained versions of foundation models should be limited to the level of associations ($\mathcal{L}_1$).

While PCH is an extremely valuable theoretical framework with wide-reaching implications, we do not think that Pearl's claim about DNNs directly follows from PCH. Training data for foundation models contains rich information about the causal structure of the world. Natural language in particular contains many descriptions of interventions and causal inferences (Fig. 1). "Passive" data, then, is not equivalent to "observational" ($\mathcal{L}_1$) data. Notably, in a more recent interview, Pearl [37] acknowledges that text does contain $\mathcal{L}_2/\mathcal{L}_3$ information. Standard PCH logic, therefore, cannot rule out the emergence of causal models in LLMs and other kinds of foundation models.

39th Conference on Neural Information Processing Systems (NeurIPS 2025).

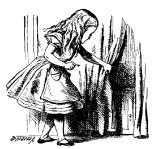
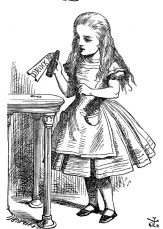
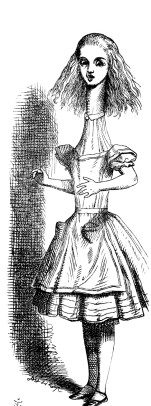

"even if my head would go through," thought poor Alice, "it would be of very little use without my shoulders".

This bottle was *not* marked "poison," so Alice ventured to taste it ... "I must be shutting up like a telescope." ... she was now only ten inches high ...

"Well, I'll eat [the cake]," said Alice, "and if it makes me grow larger, I can reach the key; and if it makes me grow smaller, I can creep under the door; so either way I'll get into the garden ...

[She] soon finished off the cake. ... "now I'm opening out like the largest telescope that ever was! Good-bye, feet!"

Figure 1: Natural language includes descriptions of interventions and causal inference. Examples from *Alice's Adventures in Wonderland* [7], illustrated by John Tenniel. The drink makes Alice shrink, while the cake causes her to grow; she also reasons counterfactually about fitting through the tiny door. LLMs may discover causal structure—the mechanisms that remain invariant under local interventions [43, 35]—and learn causal inference engines to *predict* the next token in such strings.

However, Pearl [37] and others [51] have maintained that LLMs cannot possess causal models. The underlying assumption seems to be that systems operating in a "statistical" or "model-free" mode [36]—regardless of the data they operate on—cannot give rise to genuine internal causal models. This reasoning emphasizes the nature of the training objective (prediction) while neglecting the content of the training data (which includes causal information). Even when applied to text containing causal information, the argument goes, prediction objectives can only capture statistical associations (however richly and abstractly structured they may be) rather than causal understanding that supports generalization to novel scenarios [37]. In other words, LLMs are "causal parrots" [51].

We propose an alternative hypothesis and provide an existence proof: The next-token prediction objective, at least in some contexts, may drive LLMs to acquire *real* causal models and causal reasoning capacities at the interventional ($\mathcal{L}_2$) and counterfactual ($\mathcal{L}_3$) levels. We test this hypothesis empirically in a controlled setting. We generate text (in a simple artificial language) describing interventional data and counterfactual inferences from a constrained class of linear Gaussian structural causal models (SCMs). Generated text strings fall into one of two classes (Fig. 3): (1) `DATA` strings describe the behavior of the referenced SCM under interventions; (2) `INFERENCE` strings describe counterfactual inferences, given the referenced SCM. The strings reference the underlying SCMs via arbitrary indices, and the SCM structure is *never* explicitly provided. We train a GPT-style transformer model to predict the next token in these strings.

To reduce next-token prediction loss in our task, the model could adopt one of two strategies: memorize the training examples, or discover the structure of the underlying SCMs and learn a more general causal inference algorithm. To distinguish these two possibilities, we devise a "generalization challenge" where the trained model has to generalize to counterfactual queries about a separate set of test SCMs ($D_{\text{test}}$) for which the model has *only* been trained with interventional data, and not with any counterfactual inference strings. The only way the model might correctly complete counterfactual inference strings about these SCMs is if it had both (1) inferred the structure of the test SCMs from the interventional data and (2) learned how to perform counterfactual inference for our class of SCMs. We find that the trained model generalizes to the test SCMs, indicating it has not memorized the answers, but has instead learned a counterfactual inference engine for our class of SCMS.

In addition to demonstrating generalization of counterfactual inference behavior, we use mechanistic interpretability tools to probe the network's representations. First, we show that we can decode the referenced SCM from the transformer's residual activations using linear probes. Second, we use probes to manipulate the SCM in the "network's mind" mid-computation using gradient descent on the activations. These interventions have predictable effects on the network's output, suggesting that our probes capture a real SCM representation *used* by the model.

Existing critiques of LLMs' causal capabilities do not clearly define what it would take for an LLM to possess an internal causal model [37, 51]. We establish three pieces of evidence that our trained

network possesses internal causal models: (1) it generalizes to unseen query-structure combinations, (2) it learns decodable representations of causal structure, and (3) these representations can be causally manipulated with predictable effects on the output. Together, our results provide an existence proof of a DNN trained to predict passive streams of data that can nonetheless discover and use causal models.

## 2   Related work

Our work contributes to the broader discussion about the extent to which LLMs and other foundation models understand the world [4, 49, 28, 50]. Pearl [35, 36] and others [12, 20, 38, 44, 11] have argued that human understanding derives from building and using powerful causal abstractions of the world. Causal capacity of LLMs thus plays an important role in this larger debate about understanding.

Some have been skeptical about the causal abilities of LLMs. Referencing Pearl's hierarchy, Zečević et al. [51] argue that LLMs are "causal parrots" that occasionally answer causal questions correctly by capturing the "correlation of causal facts". We agree that LLMs may learn to recite often repeated causal claims or exploit correlations between certain words to answer causal queries. However, here we present evidence that LLMs may *also* discover true causal structure from text and learn causal inference engines that generalize to unseen structure and query combinations.

Empirical results on LLM causal capacity have been mixed [18, 52, 17]. Several datasets and benchmarks have been created to make progress on this question [15, 16, 42]. In contrast to evaluating causal abilities of pre-trained LLMs, here we consider a relatively simple causal task (discovery and inference within a constrained set of linear Gaussian SCMs) and train a small transformer from scratch with full control over the training data. We then study not only the behavior of the model, but also its *internal representations*. We are inspired by other probing and mechanistic interpretability work [33, 9, 10, 31], particularly on emergent world representations in transformers [22, 23, 32, 24], and knowledge localization/editing [26, 14].

We use the structural causal model (SCM) formalism [35] to synthesize our training data. Prior work has shown that transformers trained on synthetic data can uncover hierarchical or compositional structure. For instance, Murty et al. [29] show that transformers can learn hierarchical syntactic rules through extended training, while Lake and Baroni [19] demonstrate systematic generalization to novel combinations through meta-learning on algebraic reasoning tasks with compositional structure. SCMs are distinct in that they encode mechanistic *causal* relationships that support interventional ($\mathcal{L}_2$) and counterfactual ($\mathcal{L}_3$) reasoning within Pearl's Causal Hierarchy [2]. Training on SCM generated data allows us to put Pearl's theoretical claims about the limitations of deep neural networks [38, 36, 37] to a direct test.

It is also worth mentioning neural-causal models [34, 47, 48, 46] that replace SCM functional equations with differentiable neural components. Instead, we train standard transformers end-to-end on next-token prediction on data generated from the SCMs, demonstrating that causal reasoning can emerge from a purely language modeling objective.

Finally, our work complements a finding from Lampinen et al. [21] that RL agents can learn "causal strategies" from purely passive offline data, which the agents can then exploit by intervening at test time to uncover causal structure. Interestingly, our results suggest that purely "passive" training on next-token prediction without any interventions carried out by the model may be sufficient to discover causal structure and learn a causal inference engine.

## 3   Methods

Our setup involves: (1) generating instances from a constrained class of linear Gaussian structural causal models (SCMs) (section 3.1)); (2) generating text using those SCMs in a simple artificial language that describes interventional data and counterfactual inferences (section 3.2); (3) setting up a generalization challenge for the transformer by providing only interventional data strings for some of the SCMs during training (section 3.3); (4) training a GPT-style transformer to predict the next token in generated text (section 3.4). We then study the emergent behavior of the model and its internal representations.

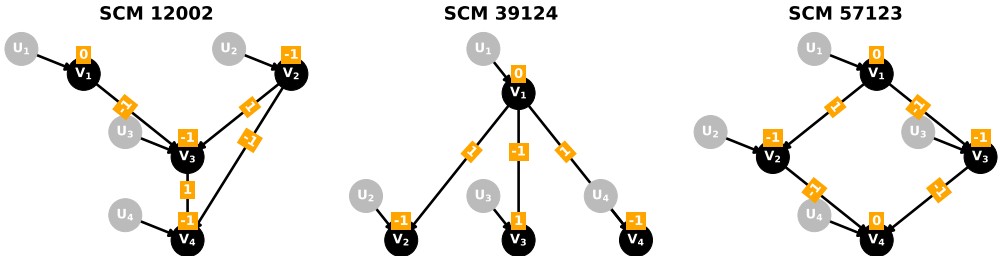

Figure 2: Some examples of the 59,049 unique SCMs. Bias weights $w_{ii}$ are displayed on top of the variables. Non-zero effect weights $w_{ij}$ are displayed on the edges between the variables.

## 3.1 Linear Gaussian structural causal models (SCMs)

We consider a constrained class of linear Gaussian structural causal models (SCMs) with 4 variables $V_1, V_2, V_3, V_4$:

$$U_j \sim \mathcal{N}(0, \sigma^2) \quad \text{with } \sigma = \sqrt{0.1} \approx 0.32 \tag{1}$$

$$V_j := U_j + w_{jj} + \sum_{i<j} w_{ij} V_i \quad \text{where } \forall i, j : w_{ij} \in \{-1, 0, 1\} \tag{2}$$

Background variables $\boldsymbol{U} = \{U_1, U_2, U_3, U_4\}$ (sometimes referred to as "exogenous" or "noise" variables) are sampled independently from a fixed Gaussian distribution. Each endogenous variable $V_j$ is a linear function of the corresponding background variable $U_j$, bias term $w_{jj}$, and a weighted combination of parent values $\sum_{i<j} w_{ij} V_i$, where $w_{ij}$ represents the effect of variable $V_i$ on $V_j$. Note that each SCM instance within our class is fully defined by a weight vector $\boldsymbol{w}$ with 10 ternary values:

$$\boldsymbol{w} = [w_{11}, w_{12}, w_{13}, w_{14}, w_{22}, w_{23}, w_{24}, w_{33}, w_{34}, w_{44}] \tag{3}$$

This weight vector $\boldsymbol{w}$ can be interpreted as the symbolic causal generative program for a particular SCM within our SCM class. Foreshadowing the results, it turns out that we can linearly decode $\boldsymbol{w}$ from the transformer's residual stream activations and manipulate it using gradient descent with predictable effects on model's output.

We generate all possible weight combinations (3 possible values for 10 weights), resulting in $3^{10} = 59,049$ unique SCM instances. Some example SCMs can be seen in Fig. 2.

Answers to counterfactual queries within the linear Gaussian SCM class can be computed analytically (details provided in appendix D.1). Efficient analytical estimates allow us to implement a text-generative model that dynamically samples queries and computes answers for each training batch.

## 3.2 Generating text (training data)

Natural language includes descriptions of interventions, outcomes under those interventions, and various causal inferences (Fig. 1). Our text generation setup using a simple artificial language is intended to emulate this aspect of natural language.

More concretely, we generate two types of strings (Fig. 3). Each string begins with a token that describes its type (`DATA` or `INFERENCE`), followed by an SCM index encoded using 4 letter tokens (e.g. `A R T Q` corresponds to SCM index 12002). Crucially, the SCM structure is never explicitly provided—the SCM indices are assigned *randomly* and, by themselves, carry no information about the underlying SCM. Therefore, the model has to discover the structure of the associated SCM from the data. We use `OBS [Vi] [value]` token sequence to represent an observation and `DO [Vi] [value]` to represent an intervention. Numbers within $[-10, 10]$ are encoded using numerical tokens with one decimal point precision (e.g. `0.3`, `-7.5`). Numbers that fall outside of that range are encoded using `-INF` and `+INF` tokens. All strings end with an `EOS` token.

`DATA` strings answer: *What happens if we intervene on X?* ($\mathcal{L}_2$ information within PCH). They provide **noisy interventional data** about the underlying SCMs. To construct a `DATA` string, we sample up to

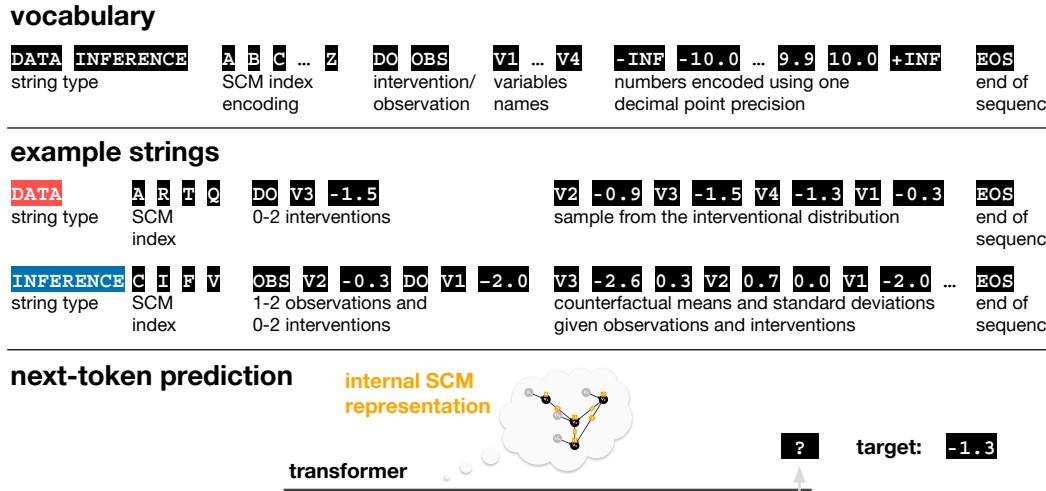

Figure 3: Top: Tokens in the vocabulary of the artificial language. Middle: Example strings from the training data. We generate two types of strings using the SCMs. DATA strings provide noisy samples from the SCM under interventions. INFERENCE strings provide examples of counterfactual inference. Bottom: We hypothesized that the GPT-style transformer may learn interpretable internal SCM representations to predict the next token.

two interventions with values drawn uniformly from $\mathcal{U}[-5, 5]$, ensuring that the intervened variables do not repeat in the same string. Given the referenced SCM and interventions, we compute the analytical interventional distribution, and *sample* one realization from that distribution. We then record the sampled variable values in a random order using [Vi] [value] token pairs. Thus, each DATA string provides one sample "rollout" from the interventional distribution—a single realization of what could happen under the intervention.

INFERENCE strings answer: *In this specific world where we observed Y, what would have happened if we had intervened on X?* ($\mathcal{L}_3$ information within PCH). They provide **examples of counterfactual inference** within our SCM class. To construct an INFERENCE string, we sample 1-2 observations (also $\sim \mathcal{U}[-5, 5]$) and 0-2 interventions. We then compute the analytical counterfactual distribution (details in Appendix D.1). In simple terms, counterfactual inference can be decomposed into three steps: "abduction", "action", and "prediction" [35]. We first condition on the observations $\boldsymbol{y}_{\text{obs}}$ and find the posterior over the background variables $p(\boldsymbol{U}|\boldsymbol{Y} = \boldsymbol{y}_{\text{obs}})$ (abduction). We then alter the factual SCM according to the interventions by fixing the intervened variables to constant values $\boldsymbol{X} := \boldsymbol{x}_{\text{int}}$ (action). Finally, we compute the counterfactual posterior distribution $P(\boldsymbol{V}_{\boldsymbol{x}_{\text{int}}}|\boldsymbol{Y} = \boldsymbol{y}_{\text{obs}})$, given the new background variables and the intervened SCM (prediction). We then use a random order to record the counterfactual *mean* and *standard deviation* for each variable using a [Vi] [mean] [std] token sequence. Unlike DATA strings which provide single samples, INFERENCE strings require computing the counterfactual posterior mean and standard deviation for a specific scenario.

Finally, we consider **two variable naming schemes** (*fixed* and *shuffled*) that map variable names (V1, V2, V3, V4) to ground truth variables ($V_1, V_2, V_3, V_4$). In the fixed naming scheme, each name refers to the same variable across SCMs: V1 $\to V_1$, V2 $\to V_2$, V3 $\to V_3$, V4 $\to V_4$. In the shuffled naming scheme, for each SCM we sample a permutation $\pi$ uniformly from all $4! = 24$ possible permutations, which then maps names to variables: Vi $\to V_{\pi(i)}$. The shuffled naming scheme is challenging since the model has to discover the name-to-variable mapping for *each* SCM—it cannot assume that, say, the variable named V1 precedes V2 in the causal order. We use the shuffled naming scheme for the main behavioral results (section 4.1) to demonstrate that the model can indeed discover the underlying SCM structures even *without* a global fixed naming scheme. However, probing models trained with shuffled names is challenging (see appendix D.2 for further explanation), so we used the fixed naming scheme for the mechanistic interpretability analyses (sections 4.2, 4.3, 4.4).

| SCM set | number of SCMs | INFERENCE strings | | DATA strings | |
|---|---|---|---|---|---|
| $D_{\text{train}}$ | 58,049 | (1) learning a **counterfactual inference engine** | ✔ | (2) building a **shared representation** | ✔ |
| $D_{\text{test}}$ | 1,000 | **generalization challenge!** | ✘ | (3) **discovering causal structure** from interventional data | ✔ |

strings seen during training? ✔ / ✘

Figure 4: Generalization challenge. To generalize to unseen INFERENCE queries for SCMs with DATA strings only ($D_{\text{test}}$) the model has to: (1) learn a counterfactual inference engine from descriptions of inferences with the $D_{\text{train}}$ SCM set, (2) build a representation for counterfactual inference and interventional data strings that exploits the shared structure, (3) discover the structure of $D_{\text{test}}$ SCMs from interventional data.

## 3.3 Generalization challenge

We wanted to distinguish between two possibilities: (1) The model memorizes answers to causal queries; (2) The model learns a causal inference engine for our SCM class. For this purpose, we devised a "generalization challenge" by randomly choosing a held-out set of 1,000 SCMs (denoted $D_{\text{test}}$) for which the model only saw DATA strings during training (Fig. 4). If the trained model can answer counterfactual queries about this test set, it means that it has (1) learned a more general counterfactual inference engine, (2) built a shared representation for interventional data and counterfactual inference, and (3) discovered the causal structure of SCMs within the $D_{\text{test}}$ set from interventional data strings. In other words, it can *compose* the learned counterfactual inference engine from $D_{\text{train}}$ strings with the discovered causal structure from interventional data strings in $D_{\text{test}}$.

## 3.4 Model architecture and training

We use the TransformerLens [30] implementation of a GPT 2-style decoder-only transformer. TransformerLens is a mechanistic interpretability library that exposes the model's internal activations for reading and editing purposes during the forward pass. We utilize these features of TransformerLens post-training when we decode SCM weights from model's residual stream activations and intervene on the SCM representation mid-computation.

Our transformer model has 12 layers, hidden size 512, 8 attention heads of size 64, MLP size 2048, GELU activation function, and "Pre-LN" type layer normalization. We use AdamW [25] with learning rate $10^{-5}$, betas [0.9, 0.999], eps $10^{-8}$, and 0.001 weight decay. We set batch size to 128 and train for different number of epochs depending on whether the variable naming scheme is fixed or shuffled (see 3.2). When variable names are *fixed*, we train for 300 epochs, reducing learning rate to $10^{-6}$ for the last 10 epochs. When variable names are *shuffled*, the model takes longer to converge, so we train for 1,500 epochs, reducing learning rate to $10^{-6}$ for the last 100 epochs. In each epoch, we draw 10 DATA and 10 INFERENCE strings per SCM from our text generative model, resulting in around 1.2 million strings per epoch. Note that there are 0 INFERENCE strings for $D_{\text{test}}$ SCMs in the training data.

## 4 Results

We show the following: (1) The trained model successfully generalizes to counterfactual queries about those SCMs that only had interventional data strings (section 4.1). (2) We can decode the SCM weights from residual stream activations within the model using linear and multi-layer perceptron (MLP) probes (section 4.2). (3) We can use these probes to manipulate the underlying SCM representation within residual activations using gradient descent with predictable effects on model's output (sections 4.3 and 4.4). Our results suggest that the next-token prediction objective drives the model to discover SCMs and learn an algorithm for counterfactual inference that generalizes within our SCM class.

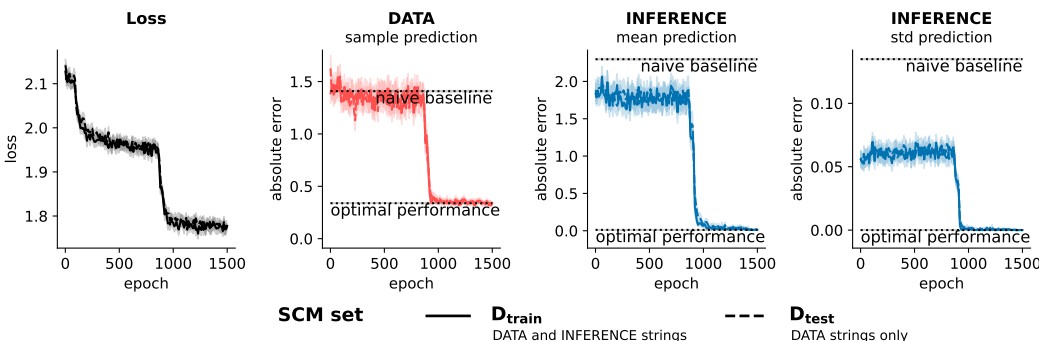

Figure 5: We track model loss and absolute error while predicting interventional `DATA` samples, counterfactual `INFERENCE` means and standard deviations for both $D_{\text{train}}$ and $D_{\text{test}}$ SCM sets. The model eventually reaches near-optimal prediction performance. Naïve baseline represents predicting the average value. Shaded regions show 95% bootstrapped confidence intervals for the mean across evaluation examples.

## 4.1 Transformer generalizes counterfactual inference to SCMs for which it has seen only interventional data

Our main behavioral test is a generalization challenge (3.3). The core intuition behind the challenge is that SCMs from the $D_{\text{test}}$ set are referenced *only* within `DATA` strings during training. If the trained model can correctly complete strings with counterfactual queries about $D_{\text{test}}$ SCMs, it means that: (1) the model has learnt a counterfactual inference engine (not just memorized the answers to counterfactual queries for the $D_{\text{train}}$ set), and (2) it has discovered the causal structure of $D_{\text{test}}$ set SCMs from noisy interventional data.

We evaluate the model by providing a causal query (string type, SCM index, observations/interventions) within its context window and letting the model predict the next-token. We then calculate the absolute error between the predicted and the ground truth values (see appendix E.1 for further details). For the main behavioral results we consider the more challenging *shuffled* variable naming scheme, where we randomly permute the names across SCMs (see 3.2). However, behavioral results also hold when variable names are *fixed* (see appendix E.3).

Fig. 5 shows the training trajectory for a few key metrics: the loss, absolute errors on interventional data samples (evaluated on `DATA` strings), and absolute errors for counterfactual means and standard deviations (evaluated on `INFERENCE` strings). The model eventually reaches near-optimal prediction performance for both $D_{\text{train}}$ and $D_{\text{test}}$ SCMs.

Fig. 6 zooms in on the last epoch and shows the trained model's performance as a function of number of interventions and observations provided in the query string. Overall, the network performs near optimally on all accuracy metrics for all conditions. Most importantly, there is no performance gap

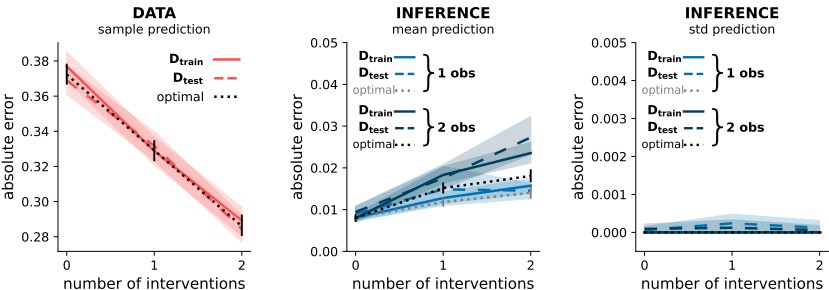

Figure 6: Trained model's errors as a function of number of interventions and observations in the query. The model performs near optimally on all prediction tasks. Importantly, performance on the $D_{\text{test}}$ SCM set does not systematically differ from $D_{\text{train}}$—the model passes the generalization challenge. Shaded regions indicate 95% bootstrapped confidence intervals for the mean across evaluation examples.

between $D_{\text{train}}$ and $D_{\text{test}}$ SCM sets—the model generalizes successfully to counterfactual queries about $D_{\text{test}}$ SCMs. We conclude that the network passes the generalization challenge, i.e. it does not simply memorize strings in the training set, but learns a counterfactual inference engine for our SCM class, discovers $D_{\text{test}}$ SCMs from strings describing interventional data, and successfully composes the discovered structure with the learnt inference engine.

## 4.2 SCM weights can be decoded from residual stream activations

In the previous section, we established that the model can generalize to unseen query-structure combinations. But what supports this behavior internally? Since the weight vector $\boldsymbol{w}$ (discussed in 3.1) fully defines an SCM within our class, we hypothesized that the model may map SCM indices to an internal representation that is closely related to this vector. To test this hypothesis, we trained "probes" [1, 3, 13] that map model's residual activations to elements of the SCM weight vector $\boldsymbol{w}$. Namely, we train a separate probe for each of the 12 transformer layers to each of the 10 SCM weights, resulting in 120 separate classifiers. We decode from residual activations at the *last SCM index position* as that position had the highest decoding accuracy—consistent with the idea that the model "pulls up" the SCM representation given the full SCM index for downstream computation.

Inspired by [23, 32], we train both linear and multi-layer perceptron (MLP) probes. Linear probes map the 512-dimensional layer $l$ residual stream activations $x_l$ directly to 3-way softmax output for each weight $w_{ij} \in \{-1, 0, 1\}$, while MLP probes include a 256-dimensional hidden layer:

$$p_{w_{ij}}^{\text{linear}}(x_l) = \text{softmax}(W^T x_l), \qquad\qquad W \in \mathbb{R}^{512 \times 3} \qquad (4)$$

$$p_{w_{ij}}^{\text{MLP}}(x_l) = \text{softmax}(W_2^T \text{ReLU}(W_1^T x_l), \qquad W_1 \in \mathbb{R}^{512 \times 256}, \quad W_2 \in \mathbb{R}^{256 \times 3} \qquad (5)$$

We split $D_{\text{train}}$ set (Fig. 4) into $D_{\text{train}}^{\text{probe}}$ (57,049 SCMs) and $D_{\text{valid}}^{\text{probe}}$ (1,000 SCMs) sets. We train the probes on $D_{\text{train}}^{\text{probe}}$, obtaining last SCM index position residual activations using `INFERENCE [SCM index]` input strings (see appendix E.4 for details on probe training). We then test probe accuracy on the $D_{\text{test}}$ SCM set (Fig. 7). SCM weights can be decoded above chance (33%) using both linear and MLP probes, suggesting that the transformer maps the arbitrary SCM index into a meaningful structured representation of the underlying SCM.

## 4.3 We can manipulate SCM representation in the residual stream

Probing results should be interpreted carefully, especially for non-linear probes [41]. Although successful decoding establishes the *presence* of information, we do not know whether or how this information is *used* by downstream computation. We hypothesized that probes do in fact recover at least parts of the model's representation of the underlying SCMs. If our hypothesis is correct, we should be able to *control* model's behavior by overwriting the SCM representation in the "network's mind" [10, 23, 27].

**Linear probe**
decoding accuracy on $D_{\text{test}}$ SCM set

| Layer | $w_{11}$ | $w_{12}$ | $w_{13}$ | $w_{14}$ | $w_{22}$ | $w_{23}$ | $w_{24}$ | $w_{33}$ | $w_{34}$ | $w_{44}$ |
|---|---|---|---|---|---|---|---|---|---|---|
| 12 | 0.96 | 0.72 | 0.66 | 0.63 | 0.60 | 0.69 | 0.67 | 0.39 | 0.68 | 0.46 |
| 11 | 0.96 | 0.70 | 0.65 | 0.62 | 0.62 | 0.69 | 0.67 | 0.41 | 0.68 | 0.46 |
| 10 | 0.98 | 0.71 | 0.67 | 0.62 | 0.62 | 0.72 | 0.66 | 0.45 | 0.68 | 0.46 |
| 9 | 0.97 | 0.72 | 0.67 | 0.61 | 0.65 | 0.72 | 0.66 | 0.48 | 0.68 | 0.45 |
| 8 | 0.98 | 0.72 | 0.67 | 0.54 | 0.70 | 0.73 | 0.65 | 0.52 | 0.68 | 0.48 |
| 7 | 1.00 | 0.76 | 0.59 | 0.43 | 0.70 | 0.74 | 0.56 | 0.56 | 0.68 | 0.50 |
| 6 | 0.98 | 0.70 | 0.48 | 0.37 | 0.65 | 0.66 | 0.49 | 0.51 | 0.65 | 0.47 |
| 5 | 0.92 | 0.58 | 0.41 | 0.37 | 0.50 | 0.54 | 0.41 | 0.41 | 0.54 | 0.44 |
| 4 | 0.75 | 0.43 | 0.36 | 0.33 | 0.41 | 0.40 | 0.37 | 0.36 | 0.43 | 0.37 |
| 3 | 0.47 | 0.36 | 0.35 | 0.31 | 0.34 | 0.34 | 0.34 | 0.34 | 0.35 | 0.35 |
| 2 | 0.36 | 0.34 | 0.35 | 0.34 | 0.34 | 0.32 | 0.33 | 0.34 | 0.31 | 0.34 |
| 1 | 0.30 | 0.34 | 0.32 | 0.36 | 0.33 | 0.35 | 0.35 | 0.31 | 0.31 | 0.33 |

SCM weight

**MLP probe**
decoding accuracy on $D_{\text{test}}$ SCM set

| Layer | $w_{11}$ | $w_{12}$ | $w_{13}$ | $w_{14}$ | $w_{22}$ | $w_{23}$ | $w_{24}$ | $w_{33}$ | $w_{34}$ | $w_{44}$ |
|---|---|---|---|---|---|---|---|---|---|---|
| 12 | 0.99 | 0.98 | 0.94 | 0.86 | 0.96 | 0.96 | 0.87 | 0.83 | 0.90 | 0.63 |
| 11 | 0.99 | 0.99 | 0.96 | 0.87 | 0.98 | 0.96 | 0.90 | 0.87 | 0.92 | 0.62 |
| 10 | 0.99 | 0.99 | 0.97 | 0.89 | 0.99 | 0.98 | 0.91 | 0.91 | 0.93 | 0.65 |
| 9 | 1.00 | 0.99 | 0.96 | 0.90 | 0.99 | 0.98 | 0.92 | 0.92 | 0.94 | 0.66 |
| 8 | 0.99 | 1.00 | 0.97 | 0.90 | 1.00 | 0.98 | 0.94 | 0.92 | 0.96 | 0.66 |
| 7 | 1.00 | 0.99 | 0.96 | 0.82 | 0.99 | 0.99 | 0.90 | 0.90 | 0.96 | 0.67 |
| 6 | 0.99 | 0.96 | 0.79 | 0.65 | 0.97 | 0.90 | 0.75 | 0.77 | 0.88 | 0.57 |
| 5 | 0.97 | 0.82 | 0.55 | 0.41 | 0.82 | 0.69 | 0.47 | 0.52 | 0.60 | 0.45 |
| 4 | 0.85 | 0.49 | 0.38 | 0.35 | 0.55 | 0.46 | 0.38 | 0.38 | 0.42 | 0.38 |
| 3 | 0.61 | 0.37 | 0.35 | 0.35 | 0.36 | 0.38 | 0.33 | 0.34 | 0.34 | 0.36 |
| 2 | 0.34 | 0.34 | 0.35 | 0.31 | 0.33 | 0.32 | 0.33 | 0.32 | 0.32 | 0.34 |
| 1 | 0.34 | 0.32 | 0.33 | 0.32 | 0.33 | 0.34 | 0.32 | 0.31 | 0.33 | 0.33 |

SCM weight

Figure 7: Average SCM weight decoding accuracy. Each cell corresponds to accuracy of the classifier that takes residual stream activations of a single layer and predicts SCM weight $w_{ij} \in \{-1, 0, 1\}$. Values in smaller font below indicate 95% bootstrapped confidence interval for the mean across the SCM set. Chance performance is 33%. By layer 5-6, most SCM weights can be decoded by linear or MLP probes above chance.

**intervening on the internal SCM representation**

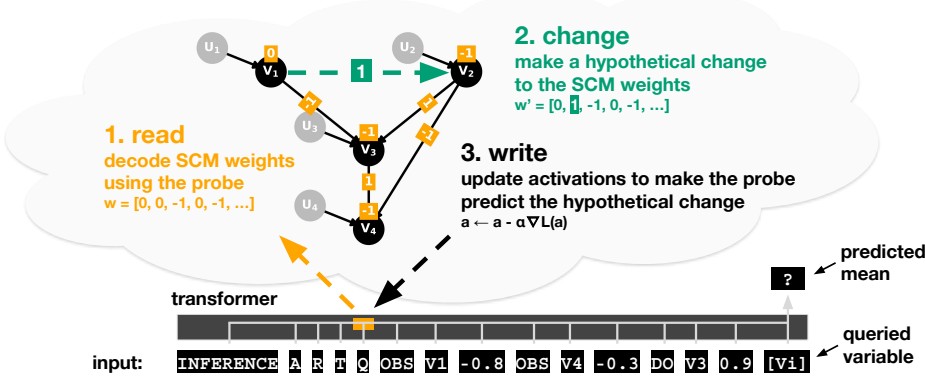

**linear probe intervention example**

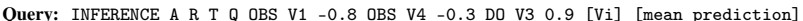

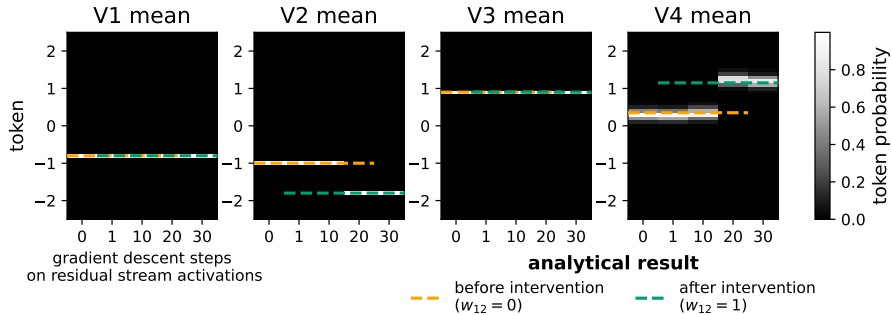

Figure 8: Top: Intervening on the internal SCM representation. Bottom: Linear probe intervention example ($w_{12} = 0 \rightarrow w_{12} = 1$). To test whether our intervention successfully modifies the representation, we: (1) specify a causal *query*, (2) compute *analytical* estimates for the counterfactual mean before ($w_{12} = 0$) and after ($w_{12} = 1$) the corresponding intervention in the ground truth SCM, and (3) check whether the post-intervention model predictions match analytical estimates. Model predictions flip for $V_2$ and $V_4$, aligning with analytical estimates.

To test our hypothesis, we use the gradient signal from the probes to intervene on the SCM representation mid-computation (Fig. 8). We can successfully manipulate the representation—the model predictions post-intervention on the residual stream align with analytical estimates for the corresponding change in the ground truth SCM, suggesting that the probes do in fact recover an actual representation of the referenced SCM. Further details on our intervention setup and an intervention example using an MLP probe can be found in appendix E.5.

## 4.4 Decoding accuracy does not imply controllability

We introduce a quantitative metric (the "intervention score") to measure how well we can *control* the network's behavior by changing its residual stream activations. Given an SCM, weight change ($w_{ij} \rightarrow w'_{ij}$), residual stream intervention at layer $l$ ($x_l \rightarrow x'_l$), and queried variable $V_{\text{queried}}$:

$$
\begin{aligned}
\text{intervention} \atop \text{score} = \frac{1}{\beta} \Bigg( \underbrace{p^{x_l \rightarrow x'_l}_{\text{model}}\big(\text{token}(E[V^{w_{ij} \rightarrow w'_{ij}}_{\text{queried}}])\big)}_{\substack{\text{probability assigned to the correct token} \\ \text{after residual stream intervention}}} - \underbrace{p_{\text{model}}\big(\text{token}(E[V^{w_{ij} \rightarrow w'_{ij}}_{\text{queried}}])\big)}_{\substack{\text{probability already assigned to the correct} \\ \text{token before residual stream intervention}}} \Bigg)
\end{aligned}
\tag{6}
$$

Note that this score can be negative when the model assigns a non-zero probability to the correct post-intervention token *before* the intervention, but the intervention on activations ($l \rightarrow l'$) lowers that probability. Moreover, the model generally hedges its bets, so we normalize the quantity in the

big brackets by the average max probability $\beta$ model assigns during normal operation ($\approx 0.89$). An average intervention score of 1 then corresponds to "perfect" interventions—the model gives *correct* answers with the same average *confidence* as during normal operation. Further details on computing intervention scores can be found in Appendix E.6.

**Linear probe**
average intervention score

| Layer | $w_{11}$ | $w_{12}$ | $w_{13}$ | $w_{14}$ | $w_{22}$ | $w_{23}$ | $w_{24}$ | $w_{33}$ | $w_{34}$ | $w_{44}$ |
|---|---|---|---|---|---|---|---|---|---|---|
| 12 | 0.00 | 0.00 | 0.00 | 0.00 | 0.00 | 0.00 | 0.00 | 0.00 | 0.00 | 0.00 |
| 11 | 0.00 | 0.00 | 0.00 | 0.00 | 0.00 | 0.00 | 0.00 | 0.00 | 0.00 | 0.00 |
| 10 | 0.00 | 0.00 | 0.00 | 0.00 | 0.00 | 0.00 | 0.00 | 0.00 | 0.00 | 0.00 |
| 9 | 0.00 | 0.00 | 0.00 | 0.00 | 0.00 | 0.00 | 0.00 | 0.00 | 0.00 | 0.00 |
| 8 | 0.00 | 0.00 | 0.00 | 0.00 | 0.00 | 0.00 | 0.00 | 0.00 | 0.00 | 0.00 |
| 7 | 0.08 | 0.02 | 0.00 | 0.00 | 0.00 | 0.03 | 0.00 | 0.01 | 0.03 | 0.02 |
| 6 | 0.29 | 0.09 | 0.06 | 0.00 | 0.06 | 0.10 | 0.08 | 0.15 | 0.09 | 0.15 |
| 5 | 0.28 | 0.11 | 0.07 | 0.02 | 0.17 | 0.09 | 0.04 | 0.18 | 0.06 | 0.10 |
| 4 | 0.26 | 0.09 | 0.07 | -0.01 | 0.19 | 0.08 | 0.01 | 0.12 | 0.01 | 0.09 |
| 3 | 0.22 | 0.06 | 0.03 | -0.04 | 0.15 | 0.02 | -0.02 | 0.08 | 0.00 | 0.07 |
| 2 | 0.17 | 0.02 | -0.01 | -0.04 | 0.11 | 0.00 | -0.06 | 0.08 | -0.04 | 0.07 |
| 1 | 0.13 | 0.01 | -0.02 | -0.04 | 0.05 | -0.01 | -0.05 | 0.06 | -0.05 | 0.06 |

**SCM weight**

**MLP probe**
average intervention score

| Layer | $w_{11}$ | $w_{12}$ | $w_{13}$ | $w_{14}$ | $w_{22}$ | $w_{23}$ | $w_{24}$ | $w_{33}$ | $w_{34}$ | $w_{44}$ |
|---|---|---|---|---|---|---|---|---|---|---|
| 12 | 0.00 | 0.00 | 0.00 | 0.00 | 0.00 | 0.00 | 0.00 | 0.00 | 0.00 | 0.00 |
| 11 | 0.00 | 0.00 | 0.00 | 0.00 | 0.00 | 0.00 | 0.00 | 0.00 | 0.00 | 0.00 |
| 10 | 0.00 | 0.00 | 0.00 | 0.00 | 0.00 | 0.00 | 0.00 | 0.00 | 0.00 | 0.00 |
| 9 | 0.00 | 0.00 | 0.00 | 0.00 | 0.00 | 0.00 | 0.00 | 0.00 | 0.00 | 0.00 |
| 8 | 0.00 | 0.00 | 0.00 | 0.00 | 0.00 | 0.00 | 0.00 | 0.00 | 0.00 | 0.00 |
| 7 | 0.02 | 0.01 | 0.02 | 0.02 | 0.00 | 0.01 | 0.01 | 0.01 | 0.01 | 0.00 |
| 6 | 0.16 | 0.08 | 0.12 | 0.10 | 0.15 | 0.10 | 0.08 | 0.16 | 0.08 | 0.14 |
| 5 | 0.24 | 0.11 | 0.11 | 0.06 | 0.20 | 0.09 | 0.05 | 0.16 | 0.06 | 0.12 |
| 4 | 0.23 | 0.06 | 0.05 | 0.01 | 0.16 | 0.06 | 0.03 | 0.09 | 0.02 | 0.09 |
| 3 | 0.21 | 0.03 | 0.04 | 0.00 | 0.10 | 0.02 | 0.00 | 0.07 | 0.02 | 0.06 |
| 2 | 0.09 | 0.02 | -0.01 | 0.00 | 0.06 | 0.01 | -0.03 | 0.06 | -0.02 | 0.07 |
| 1 | 0.03 | 0.00 | -0.01 | -0.01 | 0.02 | -0.01 | -0.02 | 0.02 | 0.00 | 0.01 |

**SCM weight**

Figure 9: Intervention scores are not coupled to decoding accuracy beyond layer 7 (compare to Fig. 7). Each cell corresponds to the average intervention score for weight $w_{ij}$ and layer $l$. Values in smaller font indicate 95% bootstrapped confidence interval for the mean across queries, queried variables $V_i$ and weight changes $w_{ij} \rightarrow w'_{ij}$.

Quantitative intervention analysis allows us to localize where the SCM representation is *used* (Fig. 9). Note that the decoding accuracy is decoupled from the intervention scores beyond layer 7—changing residual stream activations (at the last SCM index position) beyond layer 7 does not have an effect on model output. Furtermore, while the linear probe has lower overall decoding accuracy, it achieves *higher* intervention scores for some layer-SCM weight pairs (particularly, for $w_{11}$, see Fig. 14 for a contrast between linear and MLP intervention scores).

## 5   Discussion

By demonstrating that transformers can discover causal structure and learn causal inference engines solely through next-token prediction, we challenge fundamental assumptions about the limitations of neural networks trained to predict "passive" streams of data [36, 38, 37, 51]. Although the learning is passive in the sense that the model cannot perform actions in the world, the data is often rich with causal information, reflecting interventions and causal inferences performed by other agents. Even a purely predictive model may discover the underlying causal structure and learn causal inference to meet its prediction objective.

This raises broader questions that apply to all foundation models. For instance, do video generation models (e.g. OpenAI's Sora [5]) learn simulators of the world—corresponding, at least informally, to $\mathcal{L}_2$ of Pearl's Causal Hierarchy? Can we find abstract representations of physical objects and agents within their activations that we can intervene upon? If so, can we augment these models with causal reasoning capacities, so as to enable them to reason counterfactually ($\mathcal{L}_3$)? We are excited to explore these questions in future work.

The primary limitations of our work are the artificial language setup and the constrained SCM class. Our setup, while designed to emulate how natural language conveys causal information about interventions and inferences, does not capture the full richness of real-world text and causal structures. However, this constrained setting was crucial for providing a clear and unambiguous "existence proof" that statistical prediction can, in principle, give rise to causal models and reasoning. Future work could explore to what extent our results generalize to natural language descriptions of the causal scenarios and more complex causal model classes, including non-linear SCMs.

**Conclusion.** Our work provides a concrete existence proof that statistical prediction objectives can drive the emergence of causal models and causal reasoning. Through careful behavioral tests and mechanistic interpretability analyses, we demonstrated that next-token prediction can yield models that discover causal structure, build internal causal representations, and perform counterfactual inference. While our study focuses on a constrained setting, it opens new questions about the causal capabilities that may be emerging in today's foundation models trained on vast datasets.

## Acknowledgments

We thank Yushu Pan and Zhuofan Josh Ying for helpful suggestions during the early stages of the project, and Elias Bareinboim and members of the Causal Artificial Intelligence Lab at Columbia University for feedback on how to situate the work in the broader context of causality research. We also thank reviewers at NeurIPS 2025 and the Mechanistic Interpretability Workshop at NeurIPS 2025 for insightful comments that substantially improved the quality of the paper. Initial ideas for this project emerged during a book club between Eivinas Butkus and Cecilia M. Crews while reading *The Book of Why* [38].

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

# A  Code

# B  Compute Resources

Our model requires approximately 3GB VRAM and can be trained on consumer-grade hardware. Each epoch processes approximately 1.2 million examples in 10 minutes. We used one NVIDIA L40 GPU to train the final models within a university cluster using PyTorch 2.3. Evaluations, probe training, and interventional analyses were performed using a desktop machine with NVIDIA GeForce RTX 2080 Ti (10GB VRAM).

# C  Broader Impacts

**Positive impacts:** Our work provides evidence that statistical prediction objectives can drive emergence of causal models in neural networks. This advances scientific understanding of the relationship between statistical learning and causal reasoning, and puts less weight on the hypothesis that LLMs are merely "stochastic parrots" or "causal parrots" that can never, in principle, build causal abstractions through pure prediction objectives. Our mechanistic interpretability approach could inform future methods for auditing whether deployed AI systems have learned appropriate causal models for their domains.

**Potential risks:** Our results demonstrate an existence proof in a controlled setting (linear Gaussian SCMs with artificial language), which should not be misinterpreted as evidence that current LLMs have robust causal reasoning across all real-world domains. Misunderstanding our findings as a general guarantee could lead to overconfidence in deploying LLMs for high-stakes applications (medical diagnosis, policy decisions) where erroneous causal inferences could cause significant harm. Extensive domain-specific validation remains necessary before deploying AI systems that make causal judgments.

# D  Methods Appendix

## D.1  Analytic counterfactual inference in linear Gaussian SCMs

We implement analytic counterfactual inference for linear Gaussian SCMs, computing the quantity $P(\boldsymbol{V}_{\boldsymbol{x}_{\text{int}}}|\boldsymbol{Y} = \boldsymbol{y}_{\text{obs}})$, where $\boldsymbol{Y} = \boldsymbol{y}_{\text{obs}}$ are observed values in the factual world, and $\boldsymbol{x}_{\text{int}}$ represents the intervention $do(\boldsymbol{X} = \boldsymbol{x}_{\text{int}})$ in the counterfactual world. Counterfactual inference involves three steps [35]:

1. **Abduction** – finding the posterior over the background variables given observed values $p(\boldsymbol{U}|\boldsymbol{Y} = \boldsymbol{y}_{\text{obs}})$.

2. **Action** – modifying the SCM according to the intervention; in our case, this corresponds to setting a subset of the endogenous variables to constant values in the structural equations $\boldsymbol{X} := \boldsymbol{x}_{\text{int}}$ and setting incoming weights to zero for intervened variables $\mathbf{w} \to \mathbf{w}_{\boldsymbol{x}_{\text{int}}}$.

3. **Prediction** – given the posterior over background variables $\boldsymbol{U}$ and the modified SCM, computing the counterfactual distribution over the endogenous variables $\boldsymbol{V}$.

To analytically compute answers in steps (1) and (3) for a Gaussian linear SCM, it is useful to define the "total effects" matrix $T$. Let $T_{ji}$ denote the total effect of the background variable $U_i$ on the endogenous variable $V_j$, where $i, j \in \{1, \ldots, m\}$ and $m$ is the number of variables (in our case, $m = 4$). The entries of the total effects matrix $\boldsymbol{T}$ are defined recursively as follows:

1. **Base Cases:** For each target variable $V_j$ where $j = 1, \ldots, m$:
   - $T_{jj} = 1$   (Direct effect $U_j \to V_j$).
   - $T_{ji} = 0$   for all $i > j$ (No effect from later noise variables).

2. **Recursive Step:** For each target variable $V_j$ and each noise source $U_i$ where $i < j$:

$$T_{ji} = \sum_{k=i}^{j-1} T_{ki} \times w_{kj}$$

Note that $\boldsymbol{T}$ is a lower triangular matrix with ones on its diagonal. Let $\boldsymbol{b} = [w_{11}, w_{22}, w_{33}, w_{44}]^T$ denote the vector of bias terms. We can then use $\boldsymbol{T}$ and $\boldsymbol{b}$ to define the structural equations in matrix form:

$$\boldsymbol{V} = \boldsymbol{T}\boldsymbol{U} + \boldsymbol{b} \tag{7}$$

Let $\boldsymbol{y}_{\text{obs}} \in \mathbb{R}^n$ denote the vector of $n$ observed variable values, and let $\mathcal{I}_{\text{obs}} \subseteq \{1, \ldots, m\}$ be the set of indices of observed variables.

The observation matrix $\boldsymbol{H} \in \mathbb{R}^{n \times m}$ is formed by selecting the rows of the total effects matrix $\boldsymbol{T}$ corresponding to the observed variables:

$$\boldsymbol{H} = \boldsymbol{T}[\mathcal{I}_{\text{obs}}, :] \tag{8}$$

This gives us the linear relationship $\boldsymbol{y}_{\text{obs}} = \boldsymbol{H}\boldsymbol{U}$ for posterior inference. Note that bias terms here are treated as implicit in the observations rather than explicitly subtracted, which is a slight simplification.

The posterior over the background variables given the observations $p(\boldsymbol{U}|\boldsymbol{y}_{\text{obs}}) = \mathcal{N}(\boldsymbol{U}; \boldsymbol{\mu}_{\text{post}}, \boldsymbol{\Sigma}_{\text{post}})$ can be computed from the prior mean $\boldsymbol{\mu}_{\text{prior}}$ and covariance $\boldsymbol{\Sigma}_{\text{prior}}$ using standard Gaussian conditioning:

$$\boldsymbol{K} = \boldsymbol{\Sigma}_{\text{prior}} \boldsymbol{H}^T (\boldsymbol{H} \boldsymbol{\Sigma}_{\text{prior}} \boldsymbol{H}^T)^{-1} \tag{9}$$
$$\boldsymbol{\mu}_{\text{post}} = \boldsymbol{\mu}_{\text{prior}} + \boldsymbol{K}(\boldsymbol{y}_{\text{obs}} - \boldsymbol{H}\boldsymbol{\mu}_{\text{prior}}) \tag{10}$$
$$\boldsymbol{\Sigma}_{\text{post}} = \boldsymbol{\Sigma}_{\text{prior}} - \boldsymbol{K}\boldsymbol{H}\boldsymbol{\Sigma}_{\text{prior}} \tag{11}$$

where $\boldsymbol{K}$ is the Kalman gain matrix.

Let the counterfactual mean $\boldsymbol{\mu}_{\text{counter}}$ and covariance matrix $\boldsymbol{\Sigma}_{\text{counter}}$ parameterize the counterfactual distribution $P(\boldsymbol{V}_{\boldsymbol{x}_{\text{int}}}|\boldsymbol{Y} = \boldsymbol{y}_{\text{obs}}) = \mathcal{N}(\boldsymbol{V}_{\boldsymbol{x}_{\text{int}}}; \boldsymbol{\mu}_{\text{counter}}, \boldsymbol{\Sigma}_{\text{counter}})$.

To compute the counterfactual mean $\boldsymbol{\mu}_{\text{counter}}$, we compute each variable in topological order as:

$$(\boldsymbol{\mu}_{\text{counter}})_j = \begin{cases} (\boldsymbol{x}_{\text{int}})_j & \text{if variable } j \text{ is intervened upon} \\ (\boldsymbol{\mu}_{\text{post}})_j + w_{jj} + \sum_{i<j} w_{ij}(\boldsymbol{\mu}_{\text{counter}})_i & \text{otherwise} \end{cases} \tag{12}$$

To compute the covariance matrix $\boldsymbol{\Sigma}_{\text{counter}}$, we first construct a modified posterior covariance matrix $\tilde{\boldsymbol{\Sigma}}_{\text{post}}$ by setting all rows and columns corresponding to intervened variables to zero. We also compute the total effects matrix $\boldsymbol{T}_{\boldsymbol{x}_{\text{int}}}$ using the modified weight matrix $\boldsymbol{W}_{\boldsymbol{x}_{\text{int}}}$ (with incoming edges to intervened variables removed). Then:

$$\boldsymbol{\Sigma}_{\text{counter}} = \boldsymbol{T}_{\boldsymbol{x}_{\text{int}}} \tilde{\boldsymbol{\Sigma}}_{\text{post}} \boldsymbol{T}_{\boldsymbol{x}_{\text{int}}}^T \tag{13}$$

For `DATA` strings, we take one sample from the counterfactual distribution. For `INFERENCE` strings, we provide counterfactual means and standard deviations (square root of $\boldsymbol{\Sigma}_{\text{counter}}$ diagonal values).

### D.2 Motivating *fixed* variable names across SCMs for mechanistic interpretability analyses

We present behavioral results for both variable naming schemes: *shuffled* (4.1) and *fixed* (E.3). We find that the model passes our generalization challenge in both cases. However, we found that the probe decoding accuracy (mapping from activations to the SCM weights) was generally lower for the *shuffled* naming scheme, so we focused the mechanistic interpretability analyses on the case when variable names are *fixed*.

One potential reason why decoding accuracy is lower when variable names are *shuffled* is that there is a fundamental inherent ambiguity in how variables get mapped to the model's internal representation. Even if the model maintains a consistent ordered internal representation (e.g. four representational "slots" for the four variables), for certain SCMs there are multiple equivalent ways to assign variables

to these slots. This poses a challenge for our simplistic probes that assume a fixed canonical mapping between variables and internal representational positions.

Consider an SCM that has no effects between variables, i.e. all the weights between variables are zero. For such an SCM, there are $4! = 24$ equivalent ways to assign the four variables to the four internal representational slots. Since there are no causal dependencies, any assignment produces identical behavior. When variable names are shuffled randomly across SCMs, at least for certain variables within certain SCMs, the training data does not determine a unique assignment—the model is free to choose any of the equally valid internal assignments. Our probes, trained assuming a fixed mapping, struggle to decode weights consistently across these varying assignments.

In contrast, when variable names are *fixed* across all SCMs, this ambiguity is resolved—the model can use the variable names themselves as anchors to establish a consistent mapping. For instance, the model can learn to always represent the variable named `V1` in the first internal slot, `V2` in the second slot, and so on, regardless of the causal structure. This provides a canonical mapping that our probes can reliably decode.

Future work could consider more sophisticated probes (e.g. transformer-based) that can be conditioned on variable names without assuming any fixed mapping. For instance, a query for such a probe could take the form `[activations] [Vi] [Vj]`, where the target is the SCM weight (effect) between variables named `Vi` and `Vj`. Bias weights could be similarly queried by `[activations] [Vi] [Vi]`.

# E    Results Appendix

## E.1    Details on absolute error calculation

We chose mean absolute error (MAE) to measure model accuracy mostly for interpretability and robustness reasons. Unlike mean squared error (MSE), MAE is in the same units as the values being predicted and treats all prediction errors proportionally. MAE is less sensitive to occasional large outliers than MSE or RMSE (which heavily penalize large errors due to squaring), making it a more robust metric when occasional large deviations occur in the predictions.

To calculate absolute error when predicting `DATA` samples, we prompt the model with:

DATA [SCM index] [interventions] [query variable] ($\rightarrow$ [sample prediction])

and convert the highest probability next token to a numerical value. We then calculate $\text{MAE}_{\text{data}} = \frac{1}{n} \sum_{i=1}^{n} |x_i - \hat{x}_i|$, where $x_i$ is the ground-truth sampled value for example $i$, $\hat{x}_i$ is the model's predicted value, and $n$ is the total number of evaluation examples.

To calculate absolute error for `INFERENCE` mean and standard deviation, we prompt the model with:

INFERENCE [SCM index] [observations & interventions] [query variable]
($\rightarrow$ [mean prediction] [std prediction] )

and auto-regressively predict two tokens. We interpret the first as the counterfactual mean prediction and the second as the standard deviation prediction. We convert both tokens to numerical values and calculate MAE using the same formula, where ground truth values are the analytically derived counterfactual means and standard deviations.

Note that optimal absolute error is non-zero for two reasons. First, `DATA` strings provide noisy samples from interventional distributions, which cannot be predicted perfectly even with perfect knowledge of the SCM. Second, both observations/interventions in the query and predictions are encoded with one decimal point precision in the text. We compute the optimal prediction by performing exact causal inference on the encoded (one decimal point) query values, encoding the output, and comparing to the ground truth full precision values (the actual sample for `DATA` strings; the analytically computed mean/std for `INFERENCE` strings).

Note that optimal absolute error is non-zero for two reasons. First, `DATA` strings provide noisy samples of interventional distributions and it is obviously impossible to predict those perfectly. Second, for both `DATA` and `INFERENCE` strings, we encode tokens with one decimal point precision. We take that into account to compute the optimal prediction: we calculate the analytical estimate given the one decimal point precision values actually encoded in the string and compare that to the ground truth.

In Fig. 5 we present model evaluations at every 10 epochs. For each evaluated epoch and metric we sample 1,000 $D_{\text{train}}$ and 1,000 $D_{\text{test}}$ queries (where a single query is a combination of an SCM index, set of interventions, and a set of observations for INFERENCE strings). For the last epoch (Fig. 6), we sample 10,000 $D_{\text{train}}$ and 10,000 $D_{\text{test}}$ queries to obtain more precise estimates. For each query, we run the model with four different queried variables. We exclude *intervened* queried variables from the analysis since the answer is trivial—the model can achieve perfect accuracy by simply copying the intervened value from the query and predicting zero for standard deviation. Since queries contain varying numbers of interventions (0-2), excluding intervened variables leaves approximately 2.9 variables per query on average. This results in around 5,800 evaluations for each epoch (approximately 58,000 for the final epoch).

## E.2 Behavioral results hold across instances

As a robustness check, we trained two additional model instances (Fig. 10 shows all three instances for comparison). All instances successfully generalize to counterfactual queries about $D_{\text{test}}$ SCMs, reaching near-optimal performance despite somewhat idiosyncratic training trajectories.

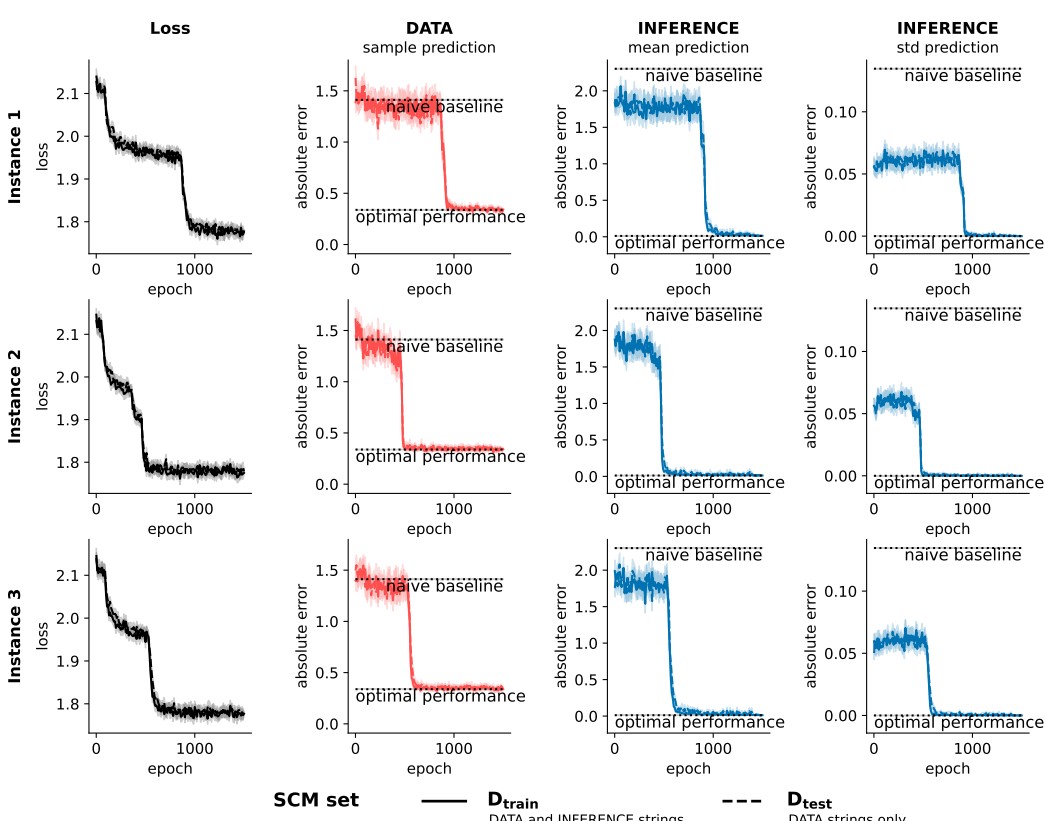

Figure 10: All three model instances pass the generalization challenge, reaching near-optimal counterfactual inference performance on $D_{\text{test}}$ SCMs (instance 1 from main text shown alongside instances 2 and 3 for comparison). Same metrics as in Fig. 5.

## E.3 Behavioral results hold when variable names are *fixed* across SCMs

Our behavioral results generalize to the somewhat simpler case when variable names are *fixed* across SCMs, which removes the variable-to-slot mapping ambiguity discussed in Appendix D.2 (Fig. 11). Note that it takes less than 300 epochs to converge to near-optimal performance in this case.

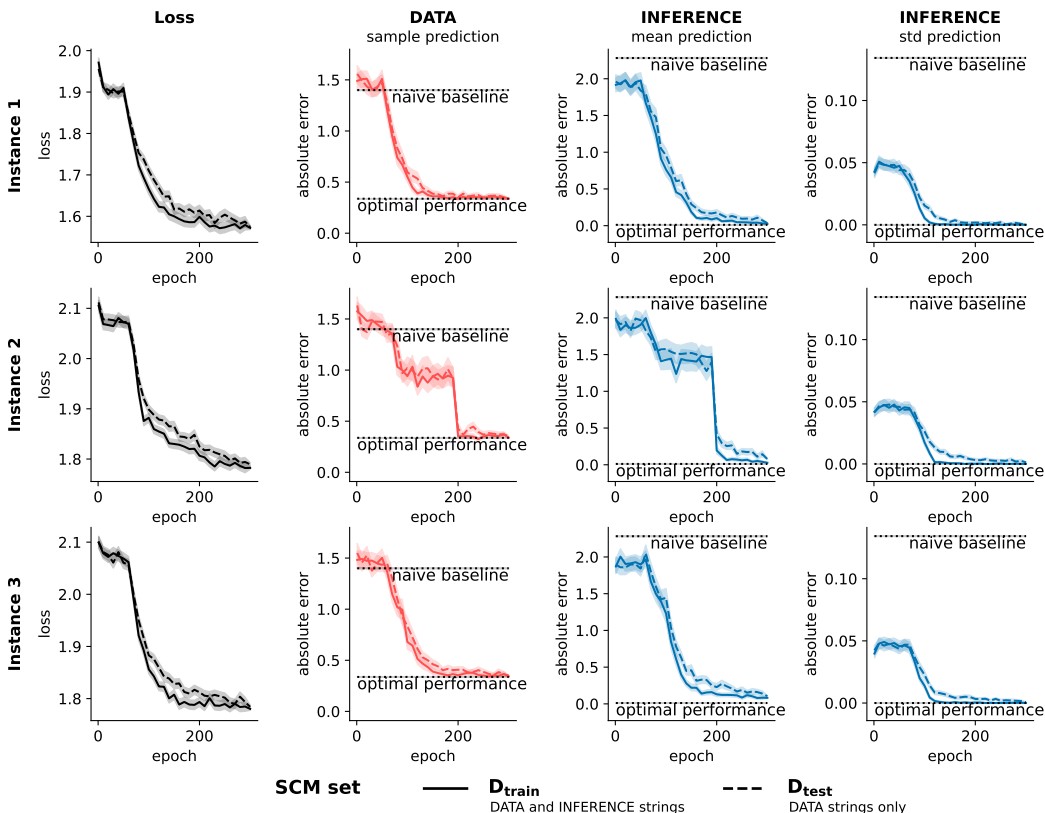

Figure 11: Training trajectories for the *fixed* variable naming scheme. The model converges to near-optimal performance faster than in the *shuffled* case. Same metrics as in Fig. 5.

### E.4 Probe training and probe accuracy on $D_{\text{valid}}^{\text{probe}}$ SCM set

We probe the residual stream *pre-activations* in the transformer model. So layer 1 activations $x_1$ correspond to token embeddings with positional embeddings, layer 2 activations $x_2$ correspond to the output of layer 1, etc.

We train both linear and MLP probes using AdamW optimizer [25] with batch size of 128, learning rate 0.001, eps $10^{-8}$, betas [0.9, 0.99], and weight decay 0.0. Due to different convergence rates, linear probes were trained for 20 epochs, and MLP probes were trained for 40 epochs. We did not observe any signs of overfitting for either of the probes based on the validation set performance.

We trained on $D_{\text{train}}^{\text{probe}}$ and tested on $D_{\text{test}}$ SCM sets (see Fig. 7). In Fig. 12 we provide probe decoding accuracy on the $D_{\text{valid}}^{\text{probe}}$ set. This is an easier generalization for the probes than the case in the main text since validation set examples come from the same $D_{\text{train}}$ set used to train the probes.

### E.5 Intervention algorithm and MLP intervention example

For all intervention experiments we only consider the INFERENCE counterfactual *mean* prediction task since that is the most challenging for the model (see Fig. 6).

Our intervention setup (Algorithm 1) is directly inspired by Li et al. [23]. The basic idea is to update the residual activations using gradient descent by minimizing the divergence between the weight predicted by the probe and the desired weight. While linear probes may allow simpler intervention schemes [32], we wanted to ensure a fair comparison between linear and MLP probes.

For the linear intervention example (Fig. 8), we intervened on layer 3 with learning rate $\alpha = 0.08$. For the MLP example (Fig. 13), we intervened on layer 5 with learning rate $\alpha = 0.08$.

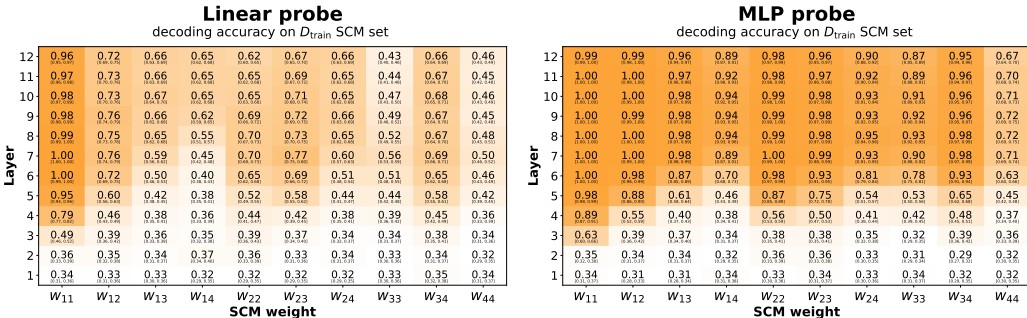

Figure 12: SCM weight decoding validation accuracy on $D_{\text{valid}}^{\text{probe}}$ set. Probe accuracy is slightly higher than for the $D_{\text{test}}$ set presented in the main text. This is expected because the probe does not need to generalize as much—both $D_{\text{train}}^{\text{probe}}$ and $D_{\text{valid}}^{\text{probe}}$ come from the same $D_{\text{train}}$ SCM set. Values in smaller font indicate 95% bootstrapped confidence interval for the mean across the SCM set.

---

**Algorithm 1** Gradient-Based Residual Stream Intervention

---

**Require:** Model $M$, probe $P$, weight index $(i, j)$, target weight $w'$, layer $\ell$, learning rate $\alpha$, steps $k$

1: $\mathbf{a} \leftarrow \mathbf{x}_\ell[:, 4, :].clone()$      ▷ Extract activations at the last SCM index position
2: $\text{Adam}(\mathbf{a}, lr = \alpha)$      ▷ Initialize Adam optimizer for $\mathbf{a}$
3: $\mathbf{y}_{\text{prev}} \leftarrow P(\mathbf{a}, \ell)$      ▷ Store initial probe predictions
4: **for** step $i = 1$ to $k$ **do**
5:      $\mathbf{y} \leftarrow P(\mathbf{a}, \ell)$      ▷ Current probe predictions
6:      $\mathcal{L}_{\text{target}} \leftarrow \text{CrossEntropy}(\mathbf{y}_{ij}, w')$      ▷ Push target weight probe prediction to $w'$
7:      $\mathcal{L}_{\text{others}} \leftarrow \text{KL}(\mathbf{y}_{\setminus ij}, \mathbf{y}_{\text{prev} \setminus ij})$      ▷ Push other probes to maintain initial predictions
8:      $\mathcal{L} \leftarrow \mathcal{L}_{\text{target}} + \mathcal{L}_{\text{others}}$
9:      Update $\mathbf{a}$ using gradient descent on $\mathcal{L}$
10: **end for**
11: $\mathbf{x}_\ell[:, 4, :] \leftarrow \mathbf{a}$
12: **return** Model output logits

---

### E.6 Intervention score details

For intervention score calculation, we only consider mean prediction in INFERENCE strings. Note also that we only consider cases where post-intervention analytic mean is different from the pre-intervention analytic mean ($E[V_k^{w_{ij} \to w'_{ij}}] \neq E[V_k^{w_{ij}}]$), i.e. we only consider variables for which the ground truth intervention "does something".

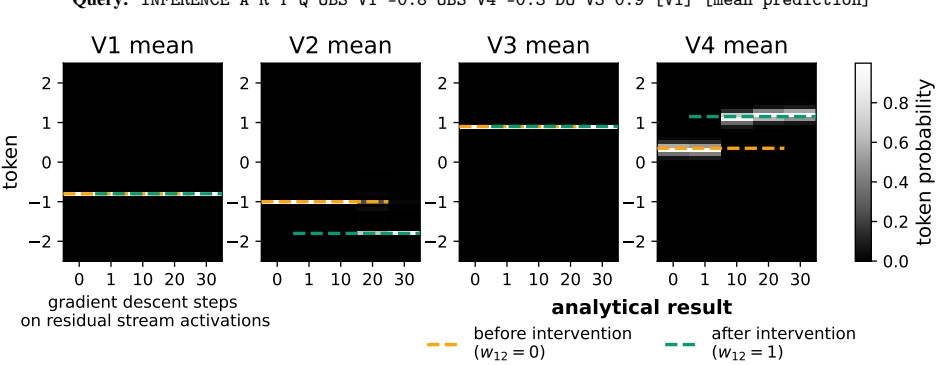

Figure 13: MLP probe intervention example ($w_{12} = 0 \to w_{12} = 1$) for the same query as in the main text (Fig. 8).

**Linear vs. MLP probe**

contrast between average intervention scores

| Layer | $w_{11}$ | $w_{12}$ | $w_{13}$ | $w_{14}$ | $w_{22}$ | $w_{23}$ | $w_{24}$ | $w_{33}$ | $w_{34}$ | $w_{44}$ |
|---|---|---|---|---|---|---|---|---|---|---|
| 12 | 0.00 | 0.00 | 0.00 | 0.00 | 0.00 | 0.00 | 0.00 | 0.00 | 0.00 | 0.00 |
| 11 | 0.00 | 0.00 | 0.00 | 0.00 | 0.00 | 0.00 | 0.00 | 0.00 | 0.00 | 0.00 |
| 10 | 0.00 | 0.00 | 0.00 | 0.00 | 0.00 | 0.00 | 0.00 | 0.00 | 0.00 | 0.00 |
| 9 | 0.00 | 0.00 | 0.00 | 0.00 | 0.00 | 0.00 | 0.00 | 0.00 | 0.00 | 0.00 |
| 8 | 0.00 | 0.00 | 0.00 | 0.00 | 0.00 | 0.00 | 0.00 | 0.00 | 0.00 | 0.00 |
| 7 | 0.06 | 0.02 | -0.01 | -0.02 | 0.00 | 0.02 | -0.01 | 0.00 | 0.02 | 0.01 |
| 6 | 0.13 | 0.02 | -0.06 | -0.10 | -0.10 | -0.01 | 0.00 | -0.01 | 0.01 | 0.00 |
| 5 | 0.04 | 0.00 | -0.03 | -0.04 | -0.03 | 0.00 | 0.00 | 0.01 | 0.00 | -0.02 |
| 4 | 0.03 | 0.03 | 0.02 | -0.02 | 0.03 | 0.02 | -0.02 | 0.03 | -0.01 | 0.00 |
| 3 | 0.01 | 0.03 | -0.01 | -0.04 | 0.05 | 0.00 | 0.00 | 0.01 | -0.01 | 0.00 |
| 2 | 0.08 | 0.00 | 0.00 | -0.04 | 0.00 | 0.05 | -0.01 | -0.03 | 0.02 | 0.00 |
| 1 | 0.10 | 0.01 | -0.01 | -0.04 | 0.04 | 0.00 | -0.03 | 0.04 | -0.04 | 0.05 |

**SCM weight**

Figure 14: Intervention score contrast between linear (blue) and MLP probes (red) (intervention score for MLP probe subtracted from the score for linear probe). While linear probes have lower decoding accuracy (Fig. 7 and 12), MLP probes do not strictly dominate them in terms of intervention scores. Values in smaller font indicate 95% bootstrapped confidence interval across queries, queried variables $V_i$ and weight changes $w_{ij} \to w'_{ij}$.

We set normalization factor $\beta = 0.89$ in eq. 6 by computing the average max probability the model assigns when predicting the mean.

We first generate a set of 200 queries that are fixed across all layers and SCM weights. We then evaluate intervention scores for each query and for each layer-SCM weight combination by changing the weight to all possible values $w_{ij} \to w'_{ij} \in \{-1, 0, 1\}$ with number of steps $k = 30$ and learning rate $\alpha = 0.08$. We found these values to result in generally good interventions qualitatively and we found quantitatively that intervention scores were quite robust to, say, changing number of steps to $k = 20$. However, in future work we should consider a much more extensive intervention hyperparameter sweep. For instance, it may be that certain layer-SCM weight combinations require a completely different number of steps $k$ or learning rate $\alpha$.

# F    Miscellaneous

## F.1    Visualizing trained model embeddings

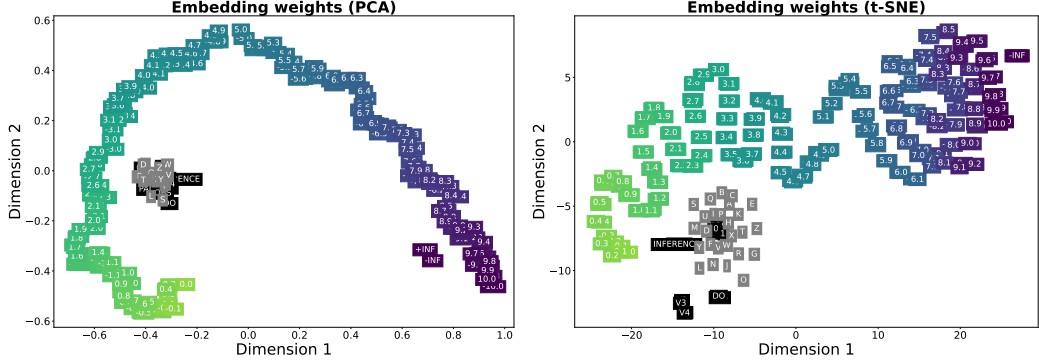

Figure 15: We visualize the trained transformer token embedding weights using PCA and t-SNE [45] (instance 1, shuffled variable names version). The model seems to learn the number line (numerical tokens color-coded based on their absolute value). The model keeps negative and positive versions of the same absolute value nearby, possibly because the prediction task requires sign flipping (when parent value is negative and weight is $-1$).

