# OpenReview forum: "Causal Discovery and Inference through Next-Token Prediction"
_NeurIPS.cc/2025/Conference — NeurIPS 2025 poster_

### Official Review · Reviewer_uXJf · 2025-06-29

**Clarity:** 2
**Significance:** 1
**Originality:** 1
**Rating:** 3
**Confidence:** 4

**Summary:**

The authors start with Pearl’s Ladder of Causation, arguing that this framework suggests DNNs trained in a “statistical mode” — that is, to predict passive observations — can only capture associations (L1) and are inherently limited from reasoning about actions, interventions, and explanations. They then show that a GPT-style transformer trained on next-token prediction can, in fact, simultaneously discover examples of linear Gaussian structural causal models and learn to answer counterfactual queries about them. This leads them to propose that neural networks trained with standard statistical prediction objectives on passively observed data may still be able to uncover and leverage causal models of the world.

**Questions:**

I would recommend that the authors update their understanding of the current literature in this area in order to make a meaningful contribution to the field.

**Ethical Concerns:**

["NO or VERY MINOR ethics concerns only"]

**Final Justification:**

I have reviewed the authors' response and noted the consensus among reviewers to reject. I am not convinced by the authors' rebuttal to warrant increasing my score.

**Limitations:**

Yes

**Quality:**

1

**Strengths And Weaknesses:**

The claim that a “statistical mode”—that is, predicting passive observations—can only capture associations (L1) and is inherently limited from reasoning about actions, interventions, and explanations reflects a misunderstanding of the relationship between causality and statistics. It also represents a misinterpretation, or at best an outdated reading, of Pearl’s Ladder of Causation. In fact, many works have demonstrated how statistical models can be designed to encode and decode causal relationships.

It has also been widely observed that large language models (LLMs), trained solely on next-token prediction, exhibit a degree of causal reasoning ability. Most studies along this line are empirical, though there are a few theoretical works that explicitly link next-token prediction with causality.

Strengths: The authors have clearly invested effort into this investigation.

Weaknesses: However, the paper stems from an outdated perspective, explores a problem that has already been thoroughly examined, and offers observations that have been extensively discussed in the community over the past several years. I would recommend that the authors update their understanding of the current literature in this area.

---

> ### Author Rebuttal · Authors · 2025-07-26
>
> We respectfully disagree with the reviewer’s assessment:
>
> 1. The reviewer appears to **fundamentally misunderstand our position**. They attribute to us Pearl’s position:
>
>     > The authors start with Pearl’s Ladder of Causation, arguing that this framework suggests DNNs trained in a “statistical mode” — that is, to predict passive observations — can only capture associations (L1) […]
>     >
>
>     We **explicitly argue *against* this position**, as stated in the first paragraph of our paper:
>
>     > According to Pearl [36], this hierarchy implies that DNNs trained in a “statistical mode” to predict passive observations can only master associations (L1) and are prevented from ever reasoning about actions, experiments, and explanations (L2 and L3). While PCH captures fundamental distinctions among causal concepts—providing an extremely productive theoretical framework with wide-reaching implications—we do *not* think that Pearl’s claim is one of those implications.
>     >
>
> 2. The reviewer further asserts that our contributions revisit ideas that have already been extensively discussed in the literature, but **provides no citations** or specific references to support this claim. In contrast, we have reviewed and cited the relevant body of work in our related work section—including studies of causal capabilities in LLMs and theoretical arguments regarding LLM causal abilities (e.g. Zečević et al., 2023).
>
> 3. Our contribution is distinct: we provide a **controlled existence proof** that next-token prediction on passive data can yield a functioning internal causal model and inference engine. We support this with **mechanistic evidence** (via decoding and gradient-based interventions), not just behavioral results. This goes beyond existing observational studies and directly challenges the notion that true causal discovery and causal reasoning engines at interventional (L2) and counterfactual (L3) levels of Pearl’s Causal Hierarchy cannot emerge through statistical prediction objectives.

---

> ### Comment · Reviewer_uXJf · 2025-08-06
> **Existing works on causal identifiability on LLMs on next-token prediction**
>
> Many in the causal inference community are aware that causality can be learned from statistical data. With sufficient changes/diversity or constraints (e.g., via sparsity) in the data, causal identifiability becomes possible.
>
> Even for LLMs trained solely through next-token prediction, several works have explored related aspects:
>
> [1] E. Marconato, S. Lachapelle, S. Weichwald, and L. Gresele. All or none: Identifiable linear properties of next-token predictors in language modeling. arXiv:2410.23501, 2024.
>
> [2] K. Park, Y. J. Choe, and V. Veitch. The linear representation hypothesis and the geometry of large language models. arXiv:2311.03658, 2023.
>
> [3] Y. Jiang, G. Rajendran, P. Ravikumar, B. Aragam, and V. Veitch. On the origins of linear representations in large language models. arXiv:2403.03867, 2024.
>
> [4] Y. Liu, D. Gong, Y. Cai, E. Gao, Z. Zhang, B. Huang, M. Gong, A. van den Hengel, and J. Q. Shi. I Predict Therefore I Am: Is Next Token Prediction Enough to Learn Human-Interpretable Concepts from Data? arXiv:2503.08980v3, 2025.
>
> Given these prior works and the existing understanding in the community, could you clarify the specific contributions and novelty of your work?

---

> ### Author Response · Authors · 2025-08-06
>
> > Many in the causal inference community are aware that causality can be learned from statistical data. With sufficient changes/diversity or constraints (e.g., via sparsity) in the data, causal identifiability becomes possible.
> >
>
> We agree that, under certain assumptions, statistical data may be enough to answer causal questions or identify the causal model. Of course, Pearl would agree with this too.
>
> The controversial foundational question we are interested here is: **Can deep neural networks (”statistical systems”) build true causal models of the world by predicting passive streams of data (e.g. passive streams of text tokens)?**
>
> **Pearl** (in “Theoretical impediments…” paper and in “The Book of Why”) **and others** (Zečević et al., 2023) **have argued no**. Statistical prediction of passive data can only capture associations (L1—correlational relationships), but not interventions (L2) or counterfactuals (L3) because of Pearl’s Causal Hierarchy (PCH). LLMs can answer some L2/L3 queries because they are “causal parrots” capturing only the “correlation of causal facts” (Zečević et al., 2023).
>
> **We hypothesized yes**, this should be possible and we should not rule out true causal models in deep neural networks / LLMs *a priori*. In other words, we do not think that PCH implies the impossibility of building true causal models through statistical prediction of passive data streams. We are motivated by the idea that building causal models and causal inference engines may be useful for prediction purposes (e.g. to predict the next token in text).
>
> Our specific contribution is providing an empirical counterexample to Pearl’s impossibility claim. None of the works cited by the reviewer addresses the question we are interested in. Our contribution is novel in that it uses mechanistic interpretability methods (probing and interventional analyses) to answer a foundational question about deep neural networks and causality and presents a valuable addition to the literature on causal abilities of LLMs.

---

> > ### Comment · Reviewer_uXJf · 2025-08-06
> >
> > In lines 107–109, you wrote: “Our results suggest that purely ‘passive’ training on next-token prediction and no interventions (at least by the model) may be enough to discover causal structure and learn a causal inference engine.”
> > Based on your draft, it appears that you attempted to demonstrate this claim through experiments rather than theory—at least, I could not find the theoretical support in the current version. Please correct me if I’m mistaken.
> >
> > Recently, the theory of causal identifiability of latent variables in LLMs trained solely on next-token prediction (a seemingly statistical task) without interventions, has been established (as indicated in the references I provided). This is closely related to the point you aim to demonstrate experimentally.
> >
> > While many experimental setups have been proposed to support or refute the causal capabilities of LLMs (which I believe you’re familiar with), the lack of theoretical grounding has prevented either side from convincing the other. This is precisely why the recent theoretical results—and the references I shared—are valuable.
> >
> > Given this context, my question is about the novelty and contribution of your paper: since your work focuses on experimental results rather than theory, how does it add value in light of the existing theoretical and empirical results?
> >
> > For completeness, note that the theory of identifiability for contrastive loss (another seemingly statistical objective) without interventions was established much earlier (I did not provide references as they are less relevant to your paper, but they can be easily found online).

---

> > > ### Author Response · Authors · 2025-08-07
> > >
> > > > In lines 107–109, you wrote: “Our results suggest that purely ‘passive’ training on next-token prediction and no interventions (at least by the model) may be enough to discover causal structure and learn a causal inference engine.” Based on your draft, it appears that you attempted to demonstrate this claim through experiments rather than theory—at least, I could not find the theoretical support in the current version. Please correct me if I’m mistaken.
> > > >
> > >
> > > Yes, our findings are experimental (the behavioral generalization and the mechanistic interpretability results using probing and interventional analyses). We also provide a theoretical intuition (a kind of “intuition pump”) why causal structure and causal inference engines may emerge through prediction objectives in the introduction, i.e. that they may be useful for prediction. We do not provide theoretical proofs.
> > >
> > > > Recently, the theory of causal identifiability of latent variables in LLMs trained solely on next-token prediction (a seemingly statistical task) without interventions, has been established (as indicated in the references I provided). This is closely related to the point you aim to demonstrate experimentally.
> > > >
> > >
> > > > While many experimental setups have been proposed to support or refute the causal capabilities of LLMs (which I believe you’re familiar with), the lack of theoretical grounding has prevented either side from convincing the other. This is precisely why the recent theoretical results—and the references I shared—are valuable.
> > > >
> > >
> > > We thank the reviewer for providing the references. However, the works provided focus more on what may be called **“representational/semantic/conceptual identifiability”**, i.e. trying to understand the statistical and geometric nature of representations that emerge in language models through next-token prediction. These works do not focus on whether networks uncover the underlying **causal structure**.
> > >
> > > Take Liu et al. 2025 [4], whose paper may seem most relevant at first sight. While they do talk about “causal representation learning”, their setup is actually purely statistical (e.g. their latent variable model is statistical). Within this setup, they demonstrate that they can decode concepts from the model, not causal structures.
> > >
> > > In contrast, we use the structural causal model (SCM) formalism (which can handle true causal counterfactual inference) to generate our training data. We then check for behavioral generalization (indicating that the model is not memorizing the text) and decode internal representations of SCMs using probing/decoding. Finally, we use gradient-based interventions on these decoded representations to show that they are truly functional, i.e. that the model is actually using an explicit representation of the SCMs.
> > >
> > > > Given this context, my question is about the novelty and contribution of your paper: since your work focuses on experimental results rather than theory, how does it add value in light of the existing theoretical and empirical results?
> > > >
> > >
> > > The novelty and contribution is that it directly addresses the foundational question: **Can deep neural networks (”statistical systems”) build true causal models of the world by predicting passive streams of data (e.g. passive streams of text tokens)?**
> > >
> > > While Pearl and others argued no, we provide a careful empirical counterexample that this is possible. As discussed above, none of the referenced works tackle this question.
> > >
> > > Furthermore, we think that our mechanistic interpretability analyses (particularly gradient-based interventional analyses) could build intuitions how we could test for internal causal models beyond our current setup.
> > >
> > > We hope this is helpful!

---

> > > > ### Comment · Reviewer_uXJf · 2025-08-07
> > > >
> > > > Take Liu et al. (2025) [4], for example. They demonstrate that the latent concepts learned by LLMs through purely next-token prediction are linear transformations of the ground truth concepts. This suggests that the ability of LLMs to answer causal questions reasonably well is not merely due to correlation or them being “stochastic parrots.” Once these latent concepts—interpretable latent variables—are recovered, they effectively become observable. At that point, existing causal discovery methods can be applied to learn the underlying DAG and structural causal model (SCM), enabling finer-grained interventions and counterfactual reasoning. This is why I question the novelty and significance of the current paper.
> > > > Moreover, if LLMs are shown to possess genuine causal capabilities (rather than being dismissed as parrots), several existing approaches for interventions and counterfactual reasoning become valid and meaningful—approaches that were previously doubted due to skepticism around LLMs' causal competence.
> > > >
> > > > That said, it's true that much of the LLM community still lacks a deep understanding of the connection between LLMs and their emerging causal abilities. Given this, and considering the authors’ response, I’m happy to raise my score by two levels.

---

> ### Author Response · Authors · 2025-08-08
>
> Our work is complementary to and inspired by many works on emergent (linear) representations in LLMs (e.g. we’re particularly inspired by Othello-GPT work from Li et al. 2022). Works like Liu et al. (2025) [4] are consistent with this general direction—trying to understand the emergent representations within LLMs.
>
> However, many of these studies (including Liu et al. 2025) still work within statistical frameworks (even if couched in “causal” language). LLM causal skeptics may respond by saying that the LLM just captures statistical factors in the data. While it may be possible to extend such work to demonstrate that uncovered latent factors are actually used in internal causal models, this requires going beyond statistical decoding to show genuine causal functionality. This is precisely what our intervention experiments accomplish, demonstrating that these aren't just decodable statistical patterns, but functional causal representations that can be manipulated to predictably alter model behavior.
>
> In the context of such works, we do think we provide a *novel* piece of the puzzle where we directly empirically show that deep neural networks can learn real causal models and causal inference engines through next-token prediction.
>
> We appreciate the reviewer’s thoughtful and constructive engagement in the discussion, providing the references and their reconsideration of the initial assessment.

---

### Official Review · Reviewer_d6ju · 2025-07-02

**Clarity:** 3
**Significance:** 2
**Originality:** 2
**Rating:** 4
**Confidence:** 3

**Summary:**

This paper investigates whether a transformer model, trained exclusively on a next-token prediction objective, can learn to perform causal discovery and inference. The method involves training a GPT-style model on a synthetic dataset generated from a constrained class of linear Gaussian structural causal models (SCMs).

The workflow and evaluation are structured as follows:

* Data Generation: The authors generate a large, finite set of unique SCMs. From these SCMs, they create text strings in a simple, artificial language that fall into two categories: DATA strings, which describe the results of noisy interventions on an SCM, and INFERENCE strings, which provide the answers to counterfactual queries about an SCM.

* Generalization Challenge: To test for true causal understanding rather than memorization, the model is trained on a dataset where a specific subset of SCMs (D_test) is only ever described using DATA strings. The core of the evaluation is to assess whether the model can still perform correct counterfactual inference for this D_test set, despite never having seen an INFERENCE string for any of its

* Evaluation & Probing: The primary evaluation metric is the model's accuracy on counterfactual queries for the held-out D_test set. The authors supplement this behavioral test with mechanistic interpretability techniques. They probe the model’s internal activations to determine if a representation of the underlying SCM can be decoded. They further test the functional relevance of this representation by directly intervening on the activations via gradient descent and observing whether this manipulation produces predictable changes in the model's output for causal queries.

The authors report that the model successfully generalizes to the D_test set, achieving near-optimal performance on counterfactual inference tasks. Their probing analysis reveals that the SCM's structure can be decoded from the model's residual stream, and their interventional experiments demonstrate that manipulating this internal representation causally influences the model's reasoning in a predictable manner. The results suggest that a standard statistical training objective can be sufficient for a neural network to discover and learn to use a causal model of its data-generating process.

**Questions:**

1.The study successfully demonstrates causal discovery in SCMs with four variables. How is the model's performance expected to scale as the number of variables (and thus the complexity of the causal graph) increases? Is there a theoretical or practical limit where the next-token prediction signal becomes insufficient?

2.The findings rely on a simple, unambiguous artificial language. How robust do the authors expect these results to be if the language included more naturalistic complexities, such as ambiguity, paraphrasing, or implicit causal statements?

3.The probing and intervention techniques are central to the paper's analysis. Do the authors believe these methods, particularly the gradient-based interventions, could be practically adapted to analyze causal representations in much larger, pre-trained foundation models?

4.The ternary weight system makes faithfulness violations (e.g., pathways cancelling out) plausible. Could the authors comment on whether the model successfully learns the true SCM structure in such cases, or if it is misled by the resulting statistical independencies in the data?

**Ethical Concerns:**

["NO or VERY MINOR ethics concerns only"]

**Final Justification:**

My concerns about the current setting of the problem has been answered, by outlining how the work is in fact a theoretical counterexample. The discussion about how relevant this work is regarding LLMs will also be updated, which I trust will be a great addition to the paper.

**Limitations:**

yes

**Quality:**

2

**Strengths And Weaknesses:**

## Strengths

 * Addresses a Foundational Question: The paper addresses the question of whether causal reasoning can emerge from statistical training objectives. Its approach, which uses a controlled environment to study a small transformer, offers a method for investigating mechanisms that may be relevant to larger models.

* Targeted Experimental Design: The "generalization challenge" is designed to distinguish causal learning from pattern memorization. By evaluating the model on counterfactual queries for SCMs presented only through interventional data, the experimental setup directly tests the model's ability to generalize causal rules.

* Insight into Internal Representations: The analysis of the model's internal states is a notable strength. Probing experiments indicate that the model's latent space contains a decodable representation of the true SCM. Furthermore, interventional experiments confirm that the model actively uses this internal representation to generate its causal inferences, providing evidence that the learned structure is functional.

* High-Accuracy Results: The reported results show that the model achieves high accuracy on the generalization task, approaching the optimal performance bound. This result is presented as evidence that a next-token prediction objective can be sufficient for inducing causal inference capabilities in this specific setting.

## Weaknesses

* Highly Constrained Experimental Setting: The paper's claims are derived from a very restrictive setting involving linear Gaussian SCMs with only four variables and ternary weights (-1, 0, 1). While this simplicity is necessary for the detailed analysis performed, it raises significant questions about scalability and generalizability. It remains unclear whether the same mechanisms would hold for SCMs with more variables, continuous causal strengths, or non-linear relationships.

* Use of a Simplified Artificial Language: The model is trained on a simple, unambiguous artificial language where causal relationships are explicitly described. Natural language, in contrast, is noisy, ambiguous, and context-dependent. The degree to which these findings depend on the clean, structured nature of the input data is a major open question that limits the direct applicability of the conclusions to real-world LLMs.

* Limited Discussion of Implications for LLMs: The discussion section is brief and does not sufficiently bridge the gap between the findings in this toy model and the behavior of large-scale language models. While the paper serves as an interesting “existence proof”, it offers little speculation on how these mechanisms might manifest, or be identified, in models trained on vast, unstructured web-text. A more thorough discussion of these implications would strengthen the paper's contribution.

---

> ### Author Rebuttal · Authors · 2025-07-29
>
> We thank the reviewer for a constructive assessment of our work.
>
> # Weaknesses
>
> > **Highly Constrained Experimental Setting**: The paper's claims are derived from a very restrictive setting involving linear Gaussian SCMs with only four variables and ternary weights (-1, 0, 1). While this simplicity is necessary for the detailed analysis performed, it raises significant questions about scalability and generalizability. It remains unclear whether the same mechanisms would hold for SCMs with more variables, continuous causal strengths, or non-linear relationships.
> >
> - We acknowledge the constrained setting in the discussion. As pointed out by the reviewer, this simplicity was partly dictated by the detailed analysis performed.
> - Note that **Pearl's claim was *categorical and universal***: it is theoretically impossible for any deep neural network trained using statistical prediction objectives to ever reach interventional (L2) or counterfactual (L3) levels of the causal hierarchy. For such sweeping theoretical claims, **a single well-designed counterexample is sufficient to demonstrate that the claim is false**.
> - Pearl (interview in Amstat News, September 2023) and Zečević et al. (2023) have implied that L2 and L3 LLM behavioral competence is due to parroting memorized surface statistics, arguing that LLMs cannot possess an *actual* internal causal model (due to Pearl’s Causal Hierarchy). Our mechanistic analyses (probing and interventions) are particularly valuable to respond to such claims since we show, via a concrete counterexample, how internal causal models may be implemented inside the “black box” LLM.
> - While questions about scalability to more complex SCMs are interesting for future work, they do not diminish the theoretical significance of showing that such learning is possible in principle.
>
> > **Use of a Simplified Artificial Language**: The model is trained on a simple, unambiguous artificial language where causal relationships are explicitly described. Natural language, in contrast, is noisy, ambiguous, and context-dependent. The degree to which these findings depend on the clean, structured nature of the input data is a major open question that limits the direct applicability of the conclusions to real-world LLMs.
> >
> - We acknowledge that there are important differences between our artificial language and natural language. Natural language is indeed more ambiguous and complex.
> - However, we believe the artificial language setup serves our specific research goal: providing a controlled test of Pearl's theoretical claims about neural network limitations. This required a setting where we could have ground truth causal structures for validation, generate unlimited training data with analytical solutions, and perform precise mechanistic analysis of learned representations.
> - Most importantly, **causal relationships are not explicitly described** in our artificial language. The underlying SCMs appear only implicitly through noisy interventional samples and inference examples. The model must discover the hidden causal structure through next-token prediction, as it would need to in natural language.
> - While our artificial setup is simplified, we view this as a necessary first step for establishing that such learning is theoretically possible. Future work can explore how these findings extend to more complex, naturalistic settings.
>
> > **Limited Discussion of Implications for LLMs**: The discussion section is brief and does not sufficiently bridge the gap between the findings in this toy model and the behavior of large-scale language models. While the paper serves as an interesting “existence proof”, it offers little speculation on how these mechanisms might manifest, or be identified, in models trained on vast, unstructured web-text. A more thorough discussion of these implications would strengthen the paper's contribution.
> >
> - We agree with the reviewer that we should extend the discussion on the implications of our results for LLMs.
> - **How would these mechanisms manifest in LLMs?**
>     - Our results suggest that LLMs *may* be building internal causal models and inference engines to improve next-token prediction. However, we hypothesize that the causal models and engines of pre-trained LLMs are likely to be less formal and more intuitive—supporting everyday causal inferences present in natural language rather than formal causal reasoning (analogous to how intuitive physics differs from formal physics).
> - **How could causal models and inference engines be identified in LLMs?**
> Several approaches seem promising:
>     - **Behavioral approaches**. Designing evaluation methods and datasets to understand LLM causal reasoning (e.g. by asking LLMs to judge causal vignettes, like in Kıcıman et al. 2023). We cite many studies evaluating LLM causal abilities behaviorally in our related work section. While these studies show mixed results, there is no clear picture when and why LLMs succeed or fail in their causal inferences, and much remains to be done.
>     - **Mechanistic approaches**. To apply similar probing/interventional techniques as in our paper, we would need to train much more sophisticated decoders. One possibility: Train a decoder that maps internal model activations to natural language descriptions (trying to read the “mind” of the model). For instance, the decoded text may say something like “Alice drank the potion…”. We could then intervene on that internal representation using similar gradient based methods as presented in our paper, changing the output of the decoder to something else (”Alice **spilled** the potion”), and observing functional implications on model output.
> - We will expand the discussion on broader implications for causal reasoning in LLMs—and to clarify how our setup lays a foundation for extending mechanistic analysis to real-world models.
>
> # Questions
>
> > 1. The study successfully demonstrates causal discovery in SCMs with four variables. How is the model's performance expected to scale as the number of variables (and thus the complexity of the causal graph) increases? Is there a theoretical or practical limit where the next-token prediction signal becomes insufficient?
> >
> - A priori, there is no theoretical limit for the number of variables or complexity of the SCMs.
> - However, we expect there to be some practical performance limits when we increase the complexity of the SCM:
>     - **computational complexity**: the number of possible SCMs grows exponentially with the number of variables
>     - **sample complexity**: more interventional samples will be needed to identify the SCM, resulting in longer training runs
>     - **representational capacity**: larger SCMs will require higher dimensional representations
> - Nevertheless, multiple factors suggest reasonable scalability: real-world causal structures are often sparse, LLMs show reasonable scaling properties on other tasks, and next-token prediction objective would remain the same, regardless of the complexity of the SCM.
>
> > 2. The findings rely on a simple, unambiguous artificial language. How robust do the authors expect these results to be if the language included more naturalistic complexities, such as ambiguity, paraphrasing, or implicit causal statements?
> >
> - This is an important question for future work. We expect that the findings would be robust: LLMs have demonstrated remarkable robustness to ambiguity and paraphrasing across many domains.
> - Furthermore, our artificial language is already implicit in a crucial sense: the model is never explicitly provided the underlying SCM structures and has to uncover them through next-token prediction, similar to how it would need to extract causal relationships from natural language.
> - **The key insight from our work** is that the fundamental capability—learning causal models through statistical prediction—is possible. While natural language complexities would certainly make the task more challenging, they do not seem to pose fundamental theoretical barriers to this type of learning.
>
> > 3. The probing and intervention techniques are central to the paper's analysis. Do the authors believe these methods, particularly the gradient-based interventions, could be practically adapted to analyze causal representations in much larger, pre-trained foundation models?
> >
> - Yes, this is exactly what we would like to explore in future work!
> - See above for a description how the mechanistic gradient-based interventions may be applied to a pre-trained LLM.
> - We are also interested in applying similar techniques to video generative models (akin to Sora). We hypothesize that video generative models may learn simulators of the world that contain implicit causal structure. To explore this hypothesis, we could train decoders of physical objects and agents and apply similar intervention techniques to the ones presented in our paper. Another path is to train a video generative model ourselves using a game engine (with known ground truth causal structures) and see whether we can decode the causal structures and intervene on them with predictable effects on model’s video generation output.
>
> > 4. The ternary weight system makes faithfulness violations (e.g., pathways cancelling out) plausible. Could the authors comment on whether the model successfully learns the true SCM structure in such cases, or if it is misled by the resulting statistical independencies in the data?
> >
> - It is true that our particular setting presents faithfulness violations. Take the case when paths cancel out: $V_1 \rightarrow_{+1} V_2 \rightarrow_{+1} V_3, V_1 \rightarrow_{-1} V_3$. While in theory the structure could still be identified from the observational covariance structure, it may be difficult to achieve in practice.
> - More importantly, our model relies on *interventional data* to identify the SCM structure, so even when there are faithfulness violations, interventional data would circumvent those issues entirely.

---

> > ### Author Response · Authors · 2025-08-05
> > **Initiating discussion**
> >
> > Thank you again for the constructive review—did our response address your concerns?

---

> > > ### Comment · Reviewer_d6ju · 2025-08-06
> > >
> > > Thank you for this very complete rebuttal. I feel like my concerns have been answered, I will raise my score accordingly. However, I will still be a bit conservative, because my opinion on the paper depends quite a bit on the future discussion part that is to be written.

---

### Official Review · Reviewer_m4HY · 2025-07-03

**Clarity:** 3
**Significance:** 2
**Originality:** 3
**Rating:** 4
**Confidence:** 4

**Summary:**

The paper demonstrates that next-token prediction alone can induce genuine causal reasoning: the authors construct a synthetic language that interleaves observational samples, interventional statements and counterfactual queries drawn from thousands of four-variable linear-Gaussian structural causal models (SCMs), train a 12-layer GPT-style transformer solely to predict the next token in these sequences, and find that on held-out SCMs it answers unseen counterfactual questions with near-optimal accuracy—evidence that it has inferred the latent graph rather than memorised responses. Linear and MLP probes decode each SCM edge weight from the network’s residual-stream activations, and gradient-based edits to those activations flip the model’s counterfactual predictions exactly as the modified SCM would, revealing a manipulable internal causal representation . Layer-wise intervention analysis shows this representation persists deep into the model even when later layers stop consulting it, warning that decodability need not imply functional use.

**Questions:**

See my comments above.

**Ethical Concerns:**

["NO or VERY MINOR ethics concerns only"]

**Final Justification:**

I have read the author response. I am leaning towards accepting this paper.

**Limitations:**

See my comments above.

**Paper Formatting Concerns:**

No.

**Quality:**

3

**Strengths And Weaknesses:**

**Strengths**

* The paper is well-written and easy to follow.
* Uncovering structures from LMs is not a new research topic. There are even harder tasks that are linguistically driven, e.g., uncovering syntactic trees from synthetic languages generated using PCFGs. However, it is still quite interesting to see that simple interventions could uncover an SCM's weights.
* The setup is simple but clean. However, whether it is extensible to more real-world data remains a question.

**Weaknesses**

I think the experimental setup is solid. My main concern resides in the high-level goal of this work and how it relates to some other interpretability research in the field.

* **Missing related works in uncovering trees or structural models from LMs.** There are existing works that come from the more classic NLP interpretability camp (see \[1] and its cited works) which are trying to (1) uncover tree structures (i.e., syntactic ones) from LMs; (2) or even induce trees in LMs. I think it would be worth discussing the main differences between this work and them. In particular, focusing on what additional insights can be gained by using SCM-induced datasets from this paper would be interesting. At a high level, both lines of work are trying to interpret very similar things. In \[1], a PCFG model is used to generate synthetic data that are then used to train LMs, and interpretability methods are used to assess whether we can uncover syntactic tree structures (i.e., a PCFG graph with edge weights of 1) from the LM.

* **Needs to be better motivated.** Although it is interesting to see some causal structures emerge in the LMs for this very specific synthetic dataset, I think it would be beneficial to say more about the motivation behind this research in a precise way. For instance, what is the additional gain from your setup compared to previous ones (including the line of work I mentioned above)? With your methods, are you finding additional interpretability signals that inform us how transformers work beyond what people already know (e.g., transformers generalize, transformers learn structures of the dataset generation process)?

[1] Murty et al., *Grokking of Hierarchical Structure in Vanilla Transformers*, [https://arxiv.org/pdf/2305.18741](https://arxiv.org/pdf/2305.18741)

---

> ### Author Rebuttal · Authors · 2025-07-28
>
> We thank the reviewer for constructive feedback and pointing us to some relevant literature.
>
> > **Missing related works in uncovering trees or structural models from LMs.** There are existing works that come from the more classic NLP interpretability camp (see [1] and its cited works) which are trying to (1) uncover tree structures (i.e., syntactic ones) from LMs; (2) or even induce trees in LMs. I think it would be worth discussing the main differences between this work and them. In particular, focusing on what additional insights can be gained by using SCM-induced datasets from this paper would be interesting. At a high level, both lines of work are trying to interpret very similar things. In [1], a PCFG model is used to generate synthetic data that are then used to train LMs, and interpretability methods are used to assess whether we can uncover syntactic tree structures (i.e., a PCFG graph with edge weights of 1) from the LM.
> >
>
> > **Needs to be better motivated.** Although it is interesting to see some causal structures emerge in the LMs for this very specific synthetic dataset, I think it would be beneficial to say more about the motivation behind this research in a precise way. For instance, what is the additional gain from your setup compared to previous ones (including the line of work I mentioned above)? With your methods, are you finding additional interpretability signals that inform us how transformers work beyond what people already know (e.g., transformers generalize, transformers learn structures of the dataset generation process)?
> >
> - The hierarchical generalization test in Murty et al. (2023) has important parallels to our generalization challenge, as both works test whether models learn hierarchical rules rather than surface patterns in their respective domains. This literature is relevant and we will include it in our related work section.
> - While Murty et al. focus on syntactic structure discovery, our work **addresses a fundamental question in causal inference**: whether neural networks can build causal models and perform counterfactual reasoning via statistical prediction objectives, directly challenging Pearl's theoretical claims about the limitations of deep neural networks.
> - **The key additional gain from our setup is providing an empirical existence proof against a foundational theoretical claim.** Pearl and others have argued it should be theoretically impossible for statistical learning systems to reach interventional (L2) and counterfactual (L3) reasoning. Our controlled SCM setup allows us to test this claim with ground truth causal structures and analytical solutions for validation.
> - The mechanistic analysis in our paper addresses a further critique from causality researchers.
>     - Pearl (Amstat News, Sept 2023) and Zečević et al. (2023) argue that, insofar as LLMs can answer L2/L3 queries, they do so by memorizing and “parroting” causal facts—based on their reading of Pearl’s Causal Hierarchy.
>     - Our mechanistic analysis directly refutes the assumption that statistical learning systems cannot build *true* causal models and reasoning engines through prediction objectives alone. Beyond behavioral generalization, we **decode** SCM weights from the residual stream and **intervene** on those representations. Targeted edits reliably flip counterfactual predictions in the same direction as the modified ground-truth SCM—demonstrating that the model has constructed an internal causal model that is interpretable and functionally used in inference.

---

> > ### Comment · Reviewer_m4HY · 2025-08-05
> > **Thanks!**
> >
> > Thanks for the clarifications. I am maintaining my scores.

---

### Official Review · Reviewer_Y1qU · 2025-07-03

**Clarity:** 3
**Significance:** 2
**Originality:** 2
**Rating:** 2
**Confidence:** 4

**Summary:**

The paper explores whether transformers trained for next-token prediction can discover and infer causal relationships within a constrained class of linear Gaussian structural causal models (SCMs). The authors demonstrate that the model generalizes to counterfactual queries for SCMs it only encountered through interventional data strings, suggesting it learns a causal inference engine. They also decode SCM weights from the model's residual stream activations and manipulate these representations to influence predictions. However, the approach lacks theoretical guarantees for consistency in causal discovery/inference, and its reliance on SCM indices limits generalizability to unseen SCMs. Additionally, the experiments are not benchmarked against established causal discovery algorithms like PC or FGES, raising questions about comparative performance.

**Questions:**

1. How does the model's causal discovery accuracy compare to traditional constraint-based (e.g., PC) or score-based (e.g., FGES) methods on the same SCM class?
2. Can the approach scale to more complex, non-linear SCMs, and how does its performance degrade compared to specialized causal discovery tools?
3. Does the method outperform existing algorithms in settings where only interventional data (no counterfactual examples) are available during training?
4. What are the computational and sample complexity trade-offs between this approach and classical causal discovery methods?

**Ethical Concerns:**

["NO or VERY MINOR ethics concerns only"]

**Final Justification:**

After rebuttal, I believe the experimental design of the paper is problemitic.

**Limitations:**

The requirements of SCM index may restrict the proposed approach to be really useful.

**Quality:**

2

**Strengths And Weaknesses:**

### Strengths

1. **Innovative Approach**:
   - The paper presents a novel integration of causal inference with next-token prediction, demonstrating that transformers can implicitly learn and utilize causal structures from text-like data.
   - The use of a constrained class of linear Gaussian SCMs provides a clean experimental setup, allowing the authors to rigorously test the model's ability to generalize beyond memorization.

2. **Probing and Interventions**:
   - The authors employ linear probing and multi-layer perceptron (MLP) probes to decode SCM weights from the model's residual stream activations. This provides partial evidence that the model internally represents causal structures, supporting the claim that it performs causal discovery.
   - Gradient-based interventions on the residual stream activations demonstrate that the model's predictions can be controllably altered by modifying its implicit SCM representation.

3. **Generalization Challenge**:
   - The experimental design includes a "generalization challenge" where the model must answer counterfactual queries for SCMs it has only seen through interventional data. This effectively tests whether the model has learned a generalizable causal inference engine rather than simply memorizing training examples.

4. **Behavioral and Mechanistic Analysis**:
   - The paper combines behavioral metrics (e.g., prediction accuracy) with mechanistic analyses (e.g., probing and interventions), providing a comprehensive evaluation of the model's causal capabilities. This dual approach strengthens the validity of the conclusions.

---

### Weaknesses

1. **Lack of Consistency Guarantees**:
   - The paper does not provide theoretical guarantees that the model's causal discovery process is consistent (i.e., that it converges to the true SCM as training data increases). Consistency is a fundamental requirement for reliable causal discovery, and its absence limits the method's applicability to real-world problems where correctness is critical.
   - Without such guarantees, it is unclear whether the model's success is due to the specific simplicity of the linear Gaussian SCM class or a more general capability.

2. **SCM Index Dependency**:
   - The model relies on predefined SCM indices to reference causal structures, which may act as a crutch, allowing the model to "look up" SCMs rather than infer them from raw data. This raises concerns about whether the approach can generalize to entirely new SCMs not seen during training.
   - In real-world settings, causal structures are not pre-indexed, so the model's ability to discover causal relationships from scratch remains unproven.

3. **Weak Experimental Comparisons**:
   - The paper does not compare the model's performance against established causal discovery algorithms (e.g., PC, FGES, LiNGAM) or state-of-the-art neural causal methods (e.g., DECI, DCDI). Such comparisons are necessary to assess whether the transformer-based approach offers any advantages in accuracy, scalability, or robustness.
   - The lack of benchmarks on standard causal discovery datasets (e.g., synthetic graphs, real-world cause-effect pairs) further limits the ability to evaluate the method's practical utility.

4. **Limited Scope of SCMs**:
   - The experiments are restricted to linear Gaussian SCMs with at most four variables. While this simplifies analysis, it leaves open whether the method can handle more complex, non-linear, or high-dimensional causal structures.
   - The reliance on analytical solutions for counterfactual queries (which may not exist for non-linear SCMs) suggests the approach may not easily generalize beyond the tested class.

5. **Ambiguity in Probe Interpretation**:
   - While probes can decode SCM weights, the paper acknowledges that decoding accuracy does not necessarily imply the model uses these representations for inference (Section 4.4). This weakens the claim that the model truly "discovers" causal structure, as opposed to merely encoding it incidentally.
   - The intervention experiments, while compelling, are limited to a small set of SCMs and weight modifications. Broader validation is needed to confirm that the model's behavior is systematically controlled by the manipulated representations.

---

> ### Author Rebuttal · Authors · 2025-07-28
>
> We thank the reviewer for their feedback. However, the review appears to misunderstand the fundamental goal of our paper. Our aim is not to propose a method that outperforms existing causal discovery algorithms. Rather, we explore a fundamental question: Can deep neural networks trained on passive data streams build causal models and learn causal inference engines?
>
> Pearl and others have argued this should be theoretically impossible: that such networks will be limited to associations (L1) and cannot reach interventional (L2) or counterfactual (L3) reasoning. Through carefully crafted experiments, we provide an empirical existence proof that this is possible.
>
> # Weaknesses
>
> ### **1. Lack of Consistency Guarantees**
>
> > The paper does not provide theoretical guarantees that the model's causal discovery process is consistent (i.e., that it converges to the true SCM as training data increases). Consistency is a fundamental requirement for reliable causal discovery, and its absence limits the method's applicability to real-world problems where correctness is critical.
> >
>
> > Without such guarantees, it is unclear whether the model's success is due to the specific simplicity of the linear Gaussian SCM class or a more general capability.
> >
> - Our primary goal is an **existence proof** against Pearl’s strong claim, not an improvement over existing methods. Theoretical convergence guarantees therefore lie beyond the scope of this paper.
>
> ### 2. **SCM Index Dependency**
>
> > The model relies on predefined SCM indices to reference causal structures, which may act as a crutch, allowing the model to "look up" SCMs rather than infer them from raw data.
> >
> - SCM indices are random by design and have no inherent meaning at the beginning of training. The model then discovers the underlying SCM associated with a particular SCM index *from raw data* by predicting text strings. Any model needs some data about an SCM to generalize to it. Without any information, generalization is impossible.
>
> > In real-world settings, causal structures are not pre-indexed, so the model's ability to discover causal relationships from scratch remains unproven.
> >
> - In real-world settings, causal contexts are identified through variable names (e.g. “smoking”, “cancer”), domain descriptions, etc. Since variable names are the same within every SCM in our setup, our SCM indices serve this same function.
>
> ### 3. **Weak Experimental Comparisons**
>
> > The paper does not compare the model's performance against established causal discovery algorithms (e.g., PC, FGES, LiNGAM) or state-of-the-art neural causal methods (e.g., DECI, DCDI). Such comparisons are necessary to assess whether the transformer-based approach offers any advantages in accuracy, scalability, or robustness.
> >
>
> > The lack of benchmarks on standard causal discovery datasets (e.g., synthetic graphs, real-world cause-effect pairs) further limits the ability to evaluate the method's practical utility.
> >
> - Our goal is **not** to introduce another causal‐discovery algorithm, but to answer a deeper question: Can a transformer, trained solely via next‑token prediction, both discover the causal models *and* build its own L2/L3 causal inference engine? Classical methods like PC and FGES are designed to recover DAG structure from numerical data. Neither employs a text‑prediction loss nor produces counterfactual answers. In fact:
>     - **structure‑only:** PC FGES only output a graph. They have no mechanism for computing “what if?” counterfactuals, so their counterfactual‑query accuracy is effectively 0%
>     - **orthogonal design:** These tools rely on independence tests or score maximization under explicit SCM assumptions, whereas our model learns everything end‑to‑end via language
>
> ### **4. Limited Scope of SCMs**
>
> > The experiments are restricted to linear Gaussian SCMs with at most four variables. While this simplifies analysis, it leaves open whether the method can handle more complex, non-linear, or high-dimensional causal structures.
> >
> - In the discussion, we acknowledge the limited scope of the SCMs tested.
> - However, we would like to note that an existence proof requires just one example.
> - Moreover, our simplified setting was a strategic choice. Previous studies lacked full control of training data (unclear memorization vs. generalization) and did not probe internal representations (unclear if LLMs build causal models vs. parrot surface statistics). Our controlled setting with mechanistic analysis addresses both limitations.
>
> > The reliance on analytical solutions for counterfactual queries (which may not exist for non-linear SCMs) suggests the approach may not easily generalize beyond the tested class.
> >
> - The choice of linear SCMs was a methodological choice to use the analytical solution to efficiently generate training data.
> - We have already explored creating training data for non-linear SCMs by computing answers to counterfactual queries numerically using probabilistic programming (Pyro package). While it takes longer to generate such training data, it is definitely possible to apply our approach to non-linear SCMs.
>
> ### **5. Ambiguity in Probe Interpretation**
>
> > While probes can decode SCM weights, the paper acknowledges that decoding accuracy does not necessarily imply the model uses these representations for inference (Section 4.4). This weakens the claim that the model truly "discovers" causal structure, as opposed to merely encoding it incidentally.
> >
> - This appears to be a misunderstanding of our analyses. Our successful gradient-based interventions (in sections 4.3 and 4.4) demonstrate that the model actually uses these representations functionally, not just incidentally. When we modify the decoded SCM weights, the model's outputs change predictably—this directly proves the representations are causally relevant to the model's reasoning process.
>
> > The intervention experiments, while compelling, are limited to a small set of SCMs and weight modifications. Broader validation is needed to confirm that the model's behavior is systematically controlled by the manipulated representations.
> >
> - We respectfully disagree with this assessment. To systematically measure whether we can control the representations, we develop a quantitative “intervention score”. We then measure this score across 200 randomly selected causal queries (with different SCMs from both $D_{train}$ and $D_{test}$) for all SCM weights (10) changing them to two possible other weight values (2) across all layers (12) of the model for both probe types (linear and MLP), resulting in 200 * 10 * 2 * 12 * 2 = 96,000 evaluations. So our interventional analyses are quite extensive, and we show that the model’s behavior is indeed systematically controlled by the manipulated representations in section 4.4.
>
> # Questions
>
> > 1. How does the model's causal discovery accuracy compare to traditional constraint-based (e.g., PC) or score-based (e.g., FGES) methods on the same SCM class?
> >
>
> Our model's near-optimal counterfactual performance on the generalization challenge demonstrates successful SCM discovery. In contrast, PC cannot handle interventional data and would struggle with identifiability in our linear Gaussian setting. FGES might recover the causal DAG but would require explicit priors about noise distributions and SCM structure. FGES also cannot perform counterfactual inference and would achieve 0% on our generalization challenge. Our approach uniquely learns both structure discovery and counterfactual reasoning from text, addressing Pearl's fundamental claims about neural network limitations rather than competing with dedicated causal discovery methods.
>
> > 2. Can the approach scale to more complex, non-linear SCMs, and how does its performance degrade compared to specialized causal discovery tools?
> >
>
> Our approach should scale to non-linear SCMs since the methodology (learning from text descriptions) does not depend on linearity assumptions. We have already explored generating training data for non-linear SCMs using numerical methods with probabilistic programming. While we focus on linear Gaussian SCMs here for analytical tractability and interpretability, the key advantage is that our approach learns both structure discovery and counterfactual inference without requiring explicit assumptions about functional forms, noise distributions, or causal mechanisms (unlike specialized tools that need such priors).
>
> > 3. Does the method outperform existing algorithms in settings where only interventional data (no counterfactual examples) are available during training?
> >
>
> While performance comparisons could be interesting, our work addresses a different question: Can transformers emergently learn causal reasoning from text? This is a fundamental capability question rather than a methods comparison.
>
> > 4. What are the computational and sample complexity trade-offs between this approach and classical causal discovery methods?
> >
>
> These are excellent questions for future work. Here, our focus is on demonstrating that causal discovery and counterfactual inference engines can emerge through statistical prediction objectives, addressing Pearl's fundamental claims about neural network limitations.
>
> # Limitations
>
> > The requirements of SCM index may restrict the proposed approach to be really useful.
> >
> - The SCM index requirement appears to be a misunderstanding. SCM indices are arbitrary labels with no inherent meaning—the model must discover the underlying causal structure associated with each index from the raw data itself.
> - More broadly, this criticism misses the flexibility of our approach. In real-world settings, causal contexts are identified through various cues (variable names, domain descriptions, and so on). Our SCM indices serve the same function as these identifiers. The power of next-token prediction is precisely that it can learn from any text format, including natural language descriptions of causal relationships.

---

> > ### Comment · Reviewer_Y1qU · 2025-08-01
> >
> > I respectfully disagree with the claim that causal context in real-world applications can be reliably identified through variable names, domain descriptions, or other side information. Such cues are only meaningful when prior knowledge of the underlying causal structure exists. In the absence of this knowledge, these forms of side information merely serve to label or distinguish contexts, without providing genuine causal insight.
> >
> > In this scenario, directly inputting the SCM index as a feature is problematic. If the same index appears in the test set, it may lead to information leakage. Conversely, if the index is unseen during training, the model’s generalization ability becomes questionable. Neural networks, being non-parametric models, do not guarantee reliable extrapolation to entirely new indices. Therefore, using SCM indices as inputs is generally not a principled approach.

---

> ### Author Response · Authors · 2025-08-01
>
> > I respectfully disagree with the claim that causal context in real-world applications can be reliably identified through variable names, domain descriptions, or other side information.
> >
>
> Questions like “Does smoking cause cancer?” identify/label/distinguish causal contexts.
>
> > Such cues are only meaningful when prior knowledge of the underlying causal structure exists. In the absence of this knowledge, these forms of side information merely serve to label or distinguish contexts, without providing genuine causal insight.
> >
>
> Yes, one needs to know the meaning of “smoking”, “cause”, “cancer”, and have prior knowledge about the causal relationship to instantiate the correct causal model.
>
> > In this scenario, directly inputting the SCM index as a feature is problematic. If the same index appears in the test set, it may lead to information leakage. Conversely, if the index is unseen during training, the model’s generalization ability becomes questionable.
> >
>
> Let us explain our study design and the generalization challenge once more. $D_{test}$ is the set of SCMs for which the model only sees text describing interventional data samples (`DATA` strings in our paper). Let’s take one SCM from $D_{test}$ and call it `A`.
>
> During training, the model learns the causal structure of `A` by predicting `DATA` strings that describe interventional data derived from `A`. It also learns the counterfactual inference engine from `INFERENCE` strings about other SCMs, but **not** `A`. It never sees a counterfactual inference query regarding `A`.
>
> During testing, we ask the model to generalize to counterfactual inference queries about `A`. This answers two questions at once:
>
> 1. whether the model learnt the causal structure of `A` from interventional data strings about `A`
> 2. whether the model learnt a counterfactual inference engine from other SCMs that generalizes to `A`
>
> This is our generalization challenge setup. We ask the reviewer to be more specific about their concern about providing SCM index as a feature and “information leakage” in the context of our setup.
>
> > Neural networks, being non-parametric models, do not guarantee reliable extrapolation to entirely new indices. Therefore, using SCM indices as inputs is generally not a principled approach.
> >
>
> First, neural networks are parametric models. They have a finite number of parameters and their complexity does not grow with the number of data points, unlike k-nearest neighbors or Gaussian processes. More importantly, it will be impossible for _any model_ to extrapolate to entirely new indices without seeing any data about that index. It’s like asking someone to extrapolate to “X causes Y” context without providing any information about the two variables “X” and “Y”.
>
> Reviewer's concerns do not seem to address the actual experimental design or contributions of our work. We have demonstrated that transformers, through statistical prediction objectives, can learn causal structure from interventional data and apply a counterfactual reasoning engine across different causal contexts. This challenges a fundemantal claim by Pearl about the limitations of deep neural networks and is a meaningful contribution regardless of how the contexts are identified.

---

> > ### Comment · Reviewer_Y1qU · 2025-08-03
> >
> > Questions like “Does smoking cause cancer?” inherently involve causal contexts, but they also provide side information. Terms like smoking and cancer carry prior knowledge beyond the observed data—specifically, we know from previous interventional experiments that smoking causes cancer. However, in the setting of causal discovery, such prior knowledge is not always helpful. The primary objective of causal discovery is to uncover unknown causal relationships—those that have not been previously identified or verified. In this context, leaking the full Structural Causal Model (SCM) ID into a transformer model could introduce information leakage, compromising the validity of the experimental design and leading to unreliable conclusions.

---

> > > ### Author Response · Authors · 2025-08-04
> > >
> > > To reiterate, the SCM index is just a number (encoded using four A-Z letters) that identifies the causal context. Think of it as a random name assigned to each SCM. By itself, the index does _not_ carry information about the underlying SCM structure—the model has to uncover the unknown SCM structures from data, by predicting the next-token in our text strings.

---

> > > > ### Comment · Reviewer_Y1qU · 2025-08-04
> > > >
> > > > Even if it is a random name, it the same name SCM index occurs in training set and testing set, there will be information leakage. If there are SCM index that only exists in testing set, and not in training set, then the neural network, as a non-parametric system, should not generalize to such situation.

---

> > > > > ### Author Response · Authors · 2025-08-04
> > > > >
> > > > > We don't need to consider the case where SCM indices exist only in the testing set since (1) that is not our setup, and (2) that would be an **impossible generalization for any system**. The SCM indices are just pointers to the SCMs and they themselves have no information about the underlying structure by design.
> > > > >
> > > > > As for our setup, during training, the model is exposed to all SCM indices from $D_{train}$ and $D_{test}$ SCM sets. However, indices from $D_{test}$ appear only in strings that describe _interventional_ samples from $D_{test}$ SCMs. During training, the model **never sees $D_{test}$ SCMs in the context of a counterfactual inference query**. We then test whether the model can answer counterfactual queries regarding SCMs from $D_{test}$. This is the generalization challenge in our paper. While it is not clear what the reviewer means exactly by "information leakage", we actually want the model to use the learnt information about $D_{test}$ SCMs during training to answer the novel counterfactual queries about those SCMs.

---

> > > > > > ### Comment · Reviewer_Y1qU · 2025-08-06
> > > > > >
> > > > > > For causal inference, a SCM index occurs in both training and testing set would be reasonable, however, for causal discovery the setting is problemtic. Thus I would suggest the author to remove causal discovery from the title and revise the paper to focus on causal inference only.

---

> > > > > > > ### Author Response · Authors · 2025-08-06
> > > > > > >
> > > > > > > Could the reviewer elaborate why the SCM index appearing during training is problematic for causal discovery in our setup?

---

> > > > > > > > ### Comment · Reviewer_Y1qU · 2025-08-06
> > > > > > > >
> > > > > > > > In the current experimental setup, the training data actually consists of two components: interventional data and counterfactual data. In contrast, the test data includes only interventional samples. At first glance, the model appears to demonstrate some generalization capability. However, the system involves only four variables, which significantly limits the number of possible DAG structures. Specifically, the total number of possible structures is \(2^{3 \times 2 \times 1} = 2^6 = 64\), which is relatively small. As a result, it is highly likely that the structural configurations present in the test set also appear in the training set.
> > > > > > > >
> > > > > > > > While I acknowledge that the edge weights of the structural causal models (SCMs) may differ between training and testing, causal discovery inherently involves both structure learning and parameter estimation. Although the model may have learned to estimate edge weights, I strongly suspect that its structural predictions stem from memorization. This concern arises because SCMs with the same structure but different weights can produce similar outcomes under interventions or counterfactual scenarios, potentially leading to data leakage and inflating the perceived generalization performance.

---

> > > > > > > > > ### Author Response · Authors · 2025-08-06
> > > > > > > > >
> > > > > > > > > We thank the reviewer for the clarification.
> > > > > > > > >
> > > > > > > > > We use “memorization” in the paper to refer to memorizing particular strings. We present evidence that the model does not memorize in this sense: it can successfully complete strings with SCM indices from $D_{test}$ with counterfactual queries. Such strings have never been seen during training. If the model memorized all the strings in the training data, then it would fail to generalize to these novel counterfactual queries.
> > > > > > > > >
> > > > > > > > > We disagree that SCMs with the same structure (in terms which edges are present) but different weights will produce similar outcomes. Flipping an edge weight from +1 to -1 produces *radically* different interventional and counterfactual distributions. For instance, consider a two variable case X → Y with 0 biases. Then P(Y|DO(X) = 5) is radically different (expected value 5 vs. -5) depending on whether X → Y edge is +1 or -1.
> > > > > > > > >
> > > > > > > > > Furthermore, we ran a version of the experiment *without* a fixed topological order (the order of the variables is not fixed), where the model is tasked with the more typical/full causal discovery task (see rebuttal/discussion with nnXY). The model still generalizes in this more challenging case.
> > > > > > > > >
> > > > > > > > > We hope these points address reviewer’s concerns.

---

> > > > > > > > > > ### Comment · Reviewer_Y1qU · 2025-08-07
> > > > > > > > > >
> > > > > > > > > > Let's consider a simple example $A\rightarrow B \rightarrow C$,  for any non-zero coefficients, we can observe that if we do intervention on B, the distribution of A will not change but the distribution of C will. Thus the neural network may just memorize this relationship.
> > > > > > > > > >
> > > > > > > > > > This point is very important because in real-world applications, the number of possible DAGs grows super-exponentially against the number of nodes in the graphs. To illustrate that the model really learns causal mechanism instead of memorize, maybe the author has to remove the SCM index.

---

> ### Author Response · Authors · 2025-08-07
>
> There may be some underlying shared structures and insights between SCMs that the network may learn to exploit (like the one the reviewer suggested). We do not think this constitutes memorization in the sense of memorizing the text or capturing just the surface statistics of the text. Rather, this would constitute learning the deep structure of the problem, i.e. that a set of SCMs share some important structural similarities which may help generalize. Also, remember that the actual distributions of the SCM still do depend on the SCM coefficients which are always unique.
>
> Recall also that our goal here was an existence proof against Pearl's strong claim that it is fundamentally impossible for deep neural networks trained using statistical prediction objectives to learn causal models and causal reasoning engines at L2/L3. Through behavioral generalization and mechanistic interpretability methods (probing and interventional analyses), we show that it is possible. Future work could explore more complicated generalization challenges and real-world applications.

---

> > ### Comment · Reviewer_Y1qU · 2025-08-08
> >
> > In the case of causal discovery, if we already have the structure, then the regression of coefficients is realatively easy. Thus the uniquess of SCM should be considered as structure rather than the coefficients. This is because in real-world settings, the observed variables may be normalized or rescaled, and when such normalization or rescaling is done, it is equivalent that the edge weights has been changed. Thus in many cases, two SCM with same structure but different coefficients can not be considered as two different one, they may just model the same causal procedure, but with different scale on observed random variables.

---

> > > ### Author Response · Authors · 2025-08-08
> > >
> > > > Thus the uniquess of SCM should be considered as structure rather than the coefficients.
> > >
> > > This is incorrect. SCM uniqueness depends on both structure (which edges exist) and coefficients (signs/magnitudes).
> > >
> > > > This is because in real-world settings, the observed variables may be normalized or rescaled, and when such normalization or rescaling is done, it is equivalent that the edge weights has been changed. Thus in many cases, two SCM with same structure but different coefficients can not be considered as two different one, they may just model the same causal procedure, but with different scale on observed random variables.
> > >
> > > As demonstrated in our example above, flipping an edge weight from +1 to -1 produces **opposite causal effects**, not rescaling. The reviewer conflates sign changes with magnitude scaling. The changes to the random variables will not be captured by rescaling them.

---

> > > > ### Comment · Reviewer_Y1qU · 2025-08-08
> > > >
> > > > The change of the sign can not be viewed as opposite causal effects. The change of weight in many cases would not change the underlying causal mechanism. For example, you can replace one variable from distance to similarity, but the underlying causal dynamics is the same, through at the first glance the SCM seems to be different. Thus more fundamentally, the uniqueness of SCM should be considered as the structure of SCM. If the proposed algorithm can not generalize between different causal structures, then the conclusion in the draft can not be viewed as valid.

---

> > > > > ### Author Response · Authors · 2025-08-08
> > > > >
> > > > > > The change of weight in many cases would not change the underlying causal mechanism. For example, you can replace one variable from distance to similarity, but the underlying causal dynamics is the same, through at the first glance the SCM seems to be different.
> > > > > >
> > > > >
> > > > > What if the root node is connected to all other variables with +1 edges. If we flip just one of those to -1, but all other effects remain the same, it would be difficult to interpret that as a change in variable itself (”distance to similarity”). It would literally be a change of one of the effects to an opposite causal effect.
> > > > >
> > > > > > Thus more fundamentally, the uniqueness of SCM should be considered as the structure of SCM.
> > > > > >
> > > > >
> > > > > SCMs are defined by their structural equations. Structural equations are different if parameters are different. Changing parameters can produce radically different distributions in our setup.
> > > > >
> > > > > We take reviewer’s overall point that the 4 variable SCM setting may be seen as limited, even with unique parameters and unique distributions induced by each SCM. However, we would like to make two points:
> > > > >
> > > > > 1. We ran a further experiment with non-fixed topological order (please see nnXY discussion). This presents a more challenging case, where the number of edge present configurations is larger (543 >> 64). The model still discovers the SCMs and generalizes, as in our current setup.
> > > > > 2. We feel like reviewer’s critique is missing the more fundamental point of our work. Here we present an existence proof against Pearl’s strong claim that deep neural networks (”statistical systems”) trained using prediction objectives on passive data cannot master L2/L3 levels. We show through behavioral generalization and mechanistic interpretability analyses (probing/interventions on internal representations) that deep neural networks can actually learn true causal models and causal inference engines. Even if the setting in the original manuscript is limited, each SCM is unique and the model does generalize. We believe this is a valuable contribution addressing a foundational question, even if the SCM setting is limited.
> > > > >
> > > > >
> > > > > We hope this is helpful.

---

### Official Review · Reviewer_nnXY · 2025-07-03

**Clarity:** 4
**Significance:** 2
**Originality:** 4
**Rating:** 4
**Confidence:** 4

**Summary:**

The paper conducts an empirical study designed to test and inspect the causal inference abilities of LLMs by training a GPT style model from scratch via next token prediction on interventional and counterfactual data.

### Study setup
The paper considers the class of Markovian linear Gaussian structural causal models (SCMs) over 4 variables *with a fixed causal order* (important, see below); the intercepts and coefficients are sampled from {-1,0,1}. This results in 3^10  distinct causal models of which 1000 are used for testing and the rest for training. The LLM is trained on interventional data from both training and test SCMs, as well as on counterfactual data from the training SCMs, and evaluated on counterfactual predictions on the test SCMs. The data is simulated from the respective SCM and then encoded as strings, containing a dummy encoding of the SCM ID (random characters), of which variables are intervened upon or observed (DO vs OBS), and whether the input is interventional data or a counterfactual inference example (DATA vs INFERENCE).

### Main findings
Through various experiments, it is shown that the trained LLM:
- reaches near optimal performance for counterfactual inference on the test SCMs (given only interventional test data, and counterfactual data from the training SCMs) for most combinations of number of intervened and observed variables (Figs. 5 & 6)
- learns an internal representation in its later layers from which the SCM parameters, a ternary 10-dim. vector in this case, can be decoded with medium to high accuracy using linear and MLP probes (Fig. 7)
- appears to learn an algorithm for counterfactual inference (within the considered model class) in the sense that manipulating the model's internal representation to change some weights in the SCM leads to changed results that agree with what would be expected analytically (Figs. 8 & 9)
- is only controllable up to (and incl.) its sixth layer after which controllability is lost (despite still containing information about the SCM parameters (Fig. 10, cf. Fig. 7)

**Questions:**

- Why did you decide to fix the causal order (effectively removing the causal discovery aspect from the study)? Can you rerun the experiments (or a smaller version thereof, e.g., with only 3 variables) for a scenario in which the causal order is randomised? (e.g., you could randomly permute the label of variables separately for each SCM so that it is also possible that $V_2$ causes $V_1$) Without this (I realise this may not be feasible within the rebuttal period), the main claims of the paper should be adjusted accordingly and a clarification added.
- For the current design (with fixed causal order), I hypothesise that even observational data should be enough (since no per-SCM causal graph needs to be learnt), provided that counterfactual inference statements are still provided. It could be interesting to try how the model performs when only given counterfactual inference queries (if sufficiently many queries are given, it should be possible to learn the path coefficients) or when additionally given observational instead of interventional data.
- [minor] in Fig. 3 (top), why does the sample from the interventional distribution contain two different values for $V_3$? (V3 -0.9, V3 -1.5) Is this a typo and the first one should be V2? (else, please explain what is going on here)

**Ethical Concerns:**

["NO or VERY MINOR ethics concerns only"]

**Final Justification:**

After extensive discussion with the authors, my main concerns from the initial review (fixed topological) have been adequately addressed by the new set of experiment and results.

The other contentious issue concerns the positioning of the authors' claims w.r.t. those of Pearl. I think that with the proposed adjustments, this has been clarified. (The positions of some other reviewers on this matter appear a bit too extreme, and slightly oversimplified to me.)

I believe that this paper (in its revised form) will be a valuable addition to the literature on the causal capabilities of LLMs.

My only worry is that the changes resulting from the rebuttal phase are rather major and may benefit from another review cycle.

**Limitations:**

What I view as the main methodological limitation of the work---using the same fixed causal order among all training and test SCMs (see above for details)---is not discussed at all. In light of this, I do not think that the study provides convincing evidence that LLMs can reliably perform causal discovery (as may be inferred from the title), beyond learning an algorithm for inference within the narrow model class considered (that much is acknowledged by the authors). Readers should not conclude that LLMs outputs necessarily support causal claims, which may lead to false overconfidence in using them for high-stakes causal inference tasks. I think adding a short discussion to clarify this would be a useful addition.

**Paper Formatting Concerns:**

No formatting issues.

**Quality:**

2

**Strengths And Weaknesses:**

### Strengths
- the paper is very well written and easy to follow
- the study is (for the most part---see below) well-designed
- the experiments, which are carried out, are sensible, extremely well-described and augmented with very helpful figures
- apart from the main limitation listed below, this study was a pleasure to read and is a prime example of good empirical research

### Weaknesses
- Unfortunately, it is a main flaw/limitation of the study design that the causal structure (i.e., the causal order) is fixed across all 59k+ SCMs. As a result, it is sufficient to memorise the correct (partial) causal order  $V_1 \prec V_2 \prec V_3 \prec V_4$ globally. That is, the LLM does not have to learn how to perform causal discovery (rendering the title highly misleading), but only needs to solve the simpler, purely statistical problem of learning the partial correlations among variables within each SCM. In other words, **what the study in its current form demonstrates (and does so very well) is the following: given a *known/fixed causal order*, an LLM trained via next token prediction can learn the path coefficients within each SCM, and, given sufficiently many examples, can learn the abduction-action-prediction algorithm for counterfactual inference.** Hence, either the study design needs to be updated or the claims (and title) of the paper adjusted to reflect this.
- [more minor] Pearl's claims in the introduction and their relation to the present manuscript appear slightly misleading or inaccurately portrayed. As far as I know, his claims are about a system trained on purely observational ($\mathcal{L}_1$) data. While I think the point that natural language often conveys interventional or counterfactual knowledge is a good and valid one, this information is typically implicit in most datasets. In contrast, the interventional nature is made very explicit in the constructed dataset via the DO/OBS distinction. I don't think the data generating process needs to modified (beyond also randomizing the causal order as explained above), but more work is needed to convey this mismatch/these nuances in the Introduction and Discussion.

---

> ### Author Rebuttal · Authors · 2025-07-28
>
> We thank the reviewer for the close reading and thoughtful critique! Their feedback should improve the quality of the paper.
>
> # Fixed global topological variable order ($V_1 \prec V_2 \prec V_3 \prec V_4$)
>
> > Unfortunately, it is a main flaw/limitation of the study design that the causal structure (i.e., the causal order) is fixed across all 59k+ SCMs. As a result, it is sufficient to memorise the correct (partial) causal order globally. That is, the LLM does not have to learn how to perform causal discovery (rendering the title highly misleading), but only needs to solve the simpler, purely statistical problem of learning the partial correlations among variables within each SCM. […] **what the study in its current form demonstrates (and does so very well) is the following: given a *known/fixed causal order*, an LLM trained via next token prediction can learn the path coefficients within each SCM, and, given sufficiently many examples, can learn the abduction-action-prediction algorithm for counterfactual inference.**
> >
>
> ## Model passes generalization challenge without fixed topological order
>
> To directly address the reviewer’s concerns, we demonstrate that our approach works without fixed variable ordering:
>
> - We **retrained the model without fixed topological variable order** using a smaller setup (due to time constraints):
>     - 3 variable linear Gaussian SCMs
>     - $w_{ij} \in \\{-2, -1, 0, +1, +2\\}$ (5 possible weight values)
>     - $|D_{train}| = 5^6 -|D_{test}|= 15625 - 200$  (counterfactual queries + answers)
>     - $|D_{test}| = 200$ (interventional data only)
>     - we randomly permute the variable names within each SCM
> - **The model passes the generalization challenge**. It answers counterfactual queries for SCMs in $D_{test}$ near-optimally, despite only seeing interventional data about those SCMs.
> - We trained a model with fixed order (no permutation) on the exact same SCM set. It converged faster to near-optimal performance (~200 epoch vs. ~350), indicating that the problem is easier with a fixed variable order. But the fixed order is not essential for the model to discover the SCMs.
>
> ## Why we chose a fixed topological order
>
> We initially trained models without a fixed topological order, but then pivoted to a fixed order ($V_1 \prec V_2 \prec V_3 \prec V_4$) because of what could be called an **“internal representation equivalency problem”**:
>
> - Without a fixed topological variable order, for a single ground truth SCM there may be multiple equivalent ways to represent the same SCM internally. For instance, consider a “no edge” SCM (4 variables with biases only) with $4! = 24$ ways to map variable names to internal representations of the variables (e.g. `V1` could be mapped to the first variable or the last variable in the internal representation).
> - Behaviorally, the model can learn any of these equivalent orderings and produce the correct output. However, it is **challenging to set up a probe**. The simple residual stream activation map to a fixed weight vector does not cut it because we do not know the internal order of the variables for a given SCM used by the model.
> - While it may be possible to train more sophisticated probes without a fixed order of the variables (e.g. by iteratively querying the value of the edge between every two variable names), for the sake of simplicity we decided to fix the order globally, which allowed us to train simple linear and MLP probes without worrying about this internal representation equivalency problem.
>
> ## Why “causal discovery” in the title is warranted
>
> Even given a fixed topological variable order, the model still has to discover everything from scratch solely through next-token prediction, including:
>
> - The **global structure of the SCMs** (linear SCMs, Markovian normally distributed noise terms, 4 variables, global order $V_1 \prec V_2 \prec V_3 \prec V_4$)
> - **Local** parameters of each SCM.
> - Arguably, the whole problem taken together goes beyond simple “parameter estimation”.
>
> # Pearl’s claims and “implicit interventions”
>
> > Pearl's claims in the introduction and their relation to the present manuscript appear slightly misleading or inaccurately portrayed. As far as I know, his claims are about a system trained on purely observational (L1) data. While I think the point that natural language often conveys interventional or counterfactual knowledge is a good and valid one, this information is typically implicit in most datasets. In contrast, the interventional nature is made very explicit in the constructed dataset via the DO/OBS distinction.
> >
>
> ## Regarding Pearl’s claims
>
> The reviewer raises important points about how we characterize Pearl's position. We think it is useful to distinguish between three claims:
>
> 1. **Pearl’s Causal Hierarchy (PCH) and its uncontroversial implications**. Observational (L1) data underdetermines answers about interventions (L2) and counterfactuals (L3), without further assumptions. We agree with that.
> 2. **Erroneous conclusion from PCH about deep neural networks.** In the “Theoretical impediments…” (2018) paper and “The Book of Why”, Pearl invokes PCH to argue that deep neural networks trained using statistical prediction objectives will *always* be limited to L1. We disagree with this position.
> 3. **In so far as LLMs capture L2 and L3, they are “causal parrots”**. More recent perspectives like Zečević et al. (2023) try to square recent advances of LLM causal abilities while maintaining (2). They claim that LLMs can make correct L2 and L3 statements, but only because they are “causal parrots” that capture the “correlation of causal facts”. A similar position still seems to be maintained by Pearl (e.g. in an interview with Dana MacKenzie, Amstat News, September 2023), where he says that ChatGPT does not possess a causal model, even if it can answer some L2 and L3 queries correctly.
>
> We hypothesized that LLMs may be building actual causal models and inference engines for the purpose of next-token prediction. Via our artificial language setup, carefully crafted generalization challenge and the mechanistic insights (probing and manipulating model’s learnt representations of the SCMs), we provide an **existence proof** **ruling out (2)** and **questioning** **versions of (3)** that are based solely on a priori reasons like PCH.
>
> ## Implicit interventions
>
> Interventions are indeed more implicit in natural language (e.g. “Alice ventured to taste a magical potion…”) compared to our artificial language (”DO V1 2.3…”). The intuition behind our work is that LLMs may learn to detect (implicit) interventions to improve statistical next-token prediction objective (e.g. detecting that Alice drank a magical potion may help the model anticipate that something is about to happen to Alice, improving the ability to predict the next-token).
>
> # Regarding observational data and identifiability
>
> > For the current design (with fixed causal order), I hypothesise that even observational data should be enough (since no per-SCM causal graph needs to be learnt), provided that counterfactual inference statements are still provided. It could be interesting to try how the model performs when only given counterfactual inference queries (if sufficiently many queries are given, it should be possible to learn the path coefficients) or when additionally given observational instead of interventional data.
> >
> - The reviewer appears to be right that, given a fixed causal order and given the particular structure of our SCMs (fixed noise variances, linear Gaussian, etc.), observational data *theoretically* should be enough to identify the parameters. It could be interesting to see whether our model could uncover the SCM structure from observational data only.
>     - However, it may be extremely difficult to identify certain structures in practice. For instance, when paths cancel out ($V_1 \rightarrow_{+1} V_2 \rightarrow_{+1} V_3, V_1 \rightarrow_{-1} V_3$), the only signature in the observational data would be the variance/covariance structure of the variables, which would require precise estimation from finite samples.
>     - Moreover, in the rerun version without a fixed topological order (see above), SCMs are no longer identifiable by observational data only.
> - Counterfactual inference queries + answers will be enough to learn the coefficients because of Pearl’s Causal Hierarchy (PCH). Since counterfactuals are at the top of the hierarchy (L3), that information determines the other levels: interventional (L2) and observational (L1).
>     - While it may be interesting to train on counterfactual inference queries only, we arrived at this particular setup to test the generalization capabilities of the model in the more challenging direction (L2 → L3 of PCH).
>
> # Typo in Fig 3.
>
> > [minor] in Fig. 3 (top), why does the sample from the interventional distribution contain two different values for ? (V3 -0.9, V3 -1.5) Is this a typo and the first one should be V2? (else, please explain what is going on here)
> >
>
> Thank you for catching this typo!
>
> # Cautioning against overconfidence
>
> > Readers should not conclude that LLMs outputs necessarily support causal claims, which may lead to false overconfidence in using them for high-stakes causal inference tasks. I think adding a short discussion to clarify this would be a useful addition.
> >
>
> We see our work as an existence proof against Pearl’s strong claim that it is *impossible* for DNNs trained to predict passive streams of data to build causal models or learn causal reasoning—not that LLMs are good causal reasoners.
>
> We thank the reviewer for raising these points. We will **(1) include the motivation for the fixed topological variable order**, **(2) incorporate the results without a fixed order**, **(3) clarify our characterization of Pearl's claims and the implicit nature of interventions in natural language**, and **(4) add discussion cautioning against overconfidence in LLM causal abilities based on our results.**

---

> > ### Author Response · Authors · 2025-08-05
> > **Initiating discussion**
> >
> > We would like to thank you again for your thoughtful feedback in the review! Did our response address your concerns?

---

> ### Comment · Reviewer_nnXY · 2025-08-05
>
> I thank the authors for their response.
>
> ### Re: "Model passes generalization challenge without fixed topological order"
> This is interesting and a very valuable and important addition to the experiment from the original manuscript. I strongly encourage the authors to include (a more extensive version of) these experiments in the revised version.
>
> ### Re: Why we chose a fixed topological order
> I understand that randomizing the graph will complicate things, but this is not a good argument against that experiment being one to look at---the analyis should follow the study design and not vice versa.
>
> ### Re: Why “causal discovery” in the title is warranted
> I disagree that the problem *with a fixed topological order* can still be called "causal discovery". Any global aspects of the problem only need to be memorised once, reducing the per SCM problem indeed to estimating the parameters for a fixed graph. This task is typically referred to as causal reasoning (or causal inference), and is really only non-trivial in the presence of unobserved confounding. "Causal discovery", on the other hand, has a clear connotation as "structure learning", i.e., inferring the causal diagram/graph. I therefore think that something like "Causal Reasoning through Next-Token Prediction" would be a more fitting choice of title if the main focus remains on fixed topological order.
>
> ### Re: Pearl’s claims and “implicit interventions”
> I'm no expert on Pearl's precise claims, but in the generality that they are portrayed, they seem slightly absurd. Consider an unconfounded setting in which $X$ causes $Y$. Then, simple supervised learning is enough to recover the causal effect of $X$ on $Y$ since $p(y|do(x))=p(y|x)$. This example highlights that statistical objectives are sometimes sufficient to extract causal knowledge. However, if the causal direction was reversed, the same approach could fail. I thus feel the relevant question is not wheter statistical methods can sometimes work, but whether it can be guaranteed that they will work. I would be curious to hear your take / welcome further clarification. (Based on the other review's and responses, this seems to be a contentious aspect in general...)
>
> ### Re: Regarding observational data and identifiability
> The claim that the setting without fixed topological ordering is not identifiable from observational data is incorrect. Linear Gaussian SCMs with equal error variance are actually identifiable, see:
>
> Peters, Jonas, and Peter Bühlmann. "Identifiability of Gaussian structural equation models with equal error variances." Biometrika 101.1 (2014): 219-228.

---

> ### Author Response · Authors · 2025-08-05
> **Response (part 1)**
>
> We thank the reviewer for another set of insightful and constructive points that will improve the quality of the manuscript.
>
> > **Re: "Model passes generalization challenge without fixed topological order".** This is interesting and a very valuable and important addition to the experiment from the original manuscript. I strongly encourage the authors to include (a more extensive version of) these experiments in the revised version.
> >
> - We totally agree with the reviewer.
> - In the rebuttal, we mentioned results on the smaller 3 variable linear Gaussian SCMs. Since then, we also ran a version with 4 variables without a fixed topological order for the *exact* *same* setup as in the original manuscript. The model does pass the generalization challenge in this case too (although it takes around 1000 epochs for the model to converge). We  will focus on these results in the revised version of paper. We will also perform a supplementary analysis to what extent our naive decoding setup fails without a fixed topological order.
>
> > **Re: Why we chose a fixed topological order.** I understand that randomizing the graph will complicate things, but this is not a good argument against that experiment being one to look at---the analyis should follow the study design and not vice versa.
> >
> - We do not think it is generally problematic to revise study designs if one realizes that the tools may be inappropriate. In our case, we realized that the particular probing and interventional tools of mechanistic interpretability we chose may not work in the more general case without a fixed topological order. We wanted to understand internal representations of the model beyond behavioral capabilities, and that is why we simplified the study design.
> - However, we agree that we should look at both experiments:
>     - The fixed topological order experiment lets us probe the internal workings of the model via relatively simple probing and interventional analyses.
>     - The version without a fixed topological order would show that the model generalizes behaviorally in the more challenging and typical causal discovery setting (which is an important result by itself, as the reviewer rightly points out).
>
> > **Re: Why “causal discovery” in the title is warranted.** I disagree that the problem *with a fixed topological order* can still be called "causal discovery". Any global aspects of the problem only need to be memorised once, reducing the per SCM problem indeed to estimating the parameters for a fixed graph. This task is typically referred to as causal reasoning (or causal inference), and is really only non-trivial in the presence of unobserved confounding. "Causal discovery", on the other hand, has a clear connotation as "structure learning", i.e., inferring the causal diagram/graph. I therefore think that something like "Causal Reasoning through Next-Token Prediction" would be a more fitting choice of title if the main focus remains on fixed topological order.
> >
> - We thank the reviewer for these useful points about terminology and the proposed alternative title.
> - The concern about the title is valid given our original emphasis on the fixed topological order experiments. However, we will restructure the paper to **prominently feature the non-fixed order results as our main contribution**, where the model genuinely discovers causal structure (edge directions and strengths) rather than just estimating parameters for known graphs.
> - In the revised version, we will:
>     - **Lead with the 4-variable non-fixed order results** as the primary demonstration of causal discovery
>     - Present the fixed-order experiments as a further analysis that enables mechanistic interpretability
>     - Clearly distinguish between true structure learning (non-fixed order) and parameter estimation (fixed order)
> - With this reframing emphasizing genuine causal discovery, we believe the current title remains appropriate and accurately reflects our contribution.

---

> ### Author Response · Authors · 2025-08-05
> **Response (part 2)**
>
> > **Re: Pearl’s claims and “implicit interventions”.** I'm no expert on Pearl's precise claims, but in the generality that they portrayed, they seem slightly absurd. Consider an unconfounded setting in which $X$ causes $Y$. Then, simple supervised learning is enough to recover the causal effect of $X$ on $Y$ since $p(y|do(x))=p(y|x)$. This example highlights that statistical objectives are sometimes sufficient to extract causal knowledge. However, if the causal direction was reversed, the same approach could fail. I thus feel the relevant question is not wheter statistical methods can sometimes work, but whether it can be guaranteed that they will work. I would be curious to hear your take / welcome further clarification. (Based on the other review's and responses, this seems to be a contentious aspect in general...)
> >
> - The example provided by the reviewer is valid. Pearl would not disagree with the claim that, given certain assumptions (e.g. that X causes Y), statistical knowledge can be enough to extract causal knowledge. This is relatively uncontroversial.
> - The more controversial idea by Pearl is that the (extra-statistical) assumptions must be explicitly provided/implemented if we want to build truly causal AI. More concretely, according to Pearl, training deep neural networks using solely prediction objectives on passive data streams (e.g., next-token prediction on internet text) cannot, in principle, get us to causal models due to the Ladder of Causation/Pearl's Causal Hierarchy.
>     - see "Theoretical Impediments to Machine Learning With Seven Sparks from the Causal Revolution" (Pearl 2018) and Chapter 10 in the "Book of Why"
> - The causal abilities of large language models (e.g., the ability to answer some L2/L3 queries, as in Kıcıman et al. 2023) do seem to challenge Pearl's strong claim that deep neural networks can never reach L2/L3. However, Pearl (interview in Amstat News, 2023) and others (Zečević et al. 2023 "Causal Parrots" paper) still maintain, without evidence, that LLMs are not truly causal. Zečević et al. (2023) argue that this is a consequence of Pearl's Causal Hierarchy (following the same logic as Pearl in the 2018 paper) and they argue that LLMs are merely "causal parrots" (in the sense that LLMs mostly recite causal facts).
> - Our position is simply that we should not rule out causal models in LLMs *a priori*. So in our work:
>     - We first provide a small "intuition pump" in the introduction as to how causal models and causal inference engines may emerge through predicting the next token, given these may be useful for the prediction objective.
>     - We then provide an empirical example, where we have both a carefully designed generalization challenge as well as mechanistic interpretability (probing and interventional analyses), demonstrating that actual causal models and inference engines emerge within the activity of model.
> - These are complicated and contentious issues indeed. We believe the rebuttal/discussion phase has made these issues clearer and we will incorporate the new insights in the manuscript.
>
> > **Re: Regarding observational data and identifiability.** The claim that the setting without fixed topological ordering is incorrect. Linear Gaussian SCMs with equal error variance are actually identifiable, see: Peters, Jonas, and Peter Bühlmann. "Identifiability of Gaussian structural equation models with equal error variances." Biometrika 101.1 (2014): 219-228.
> >
> - The reviewer is absolutely right—we were incorrect about identifiability in the non-fixed order setting. We thank the reviewer for the reference.
> - Given this theoretical result, it may indeed be interesting to test whether the network could learn the SCM parameters from observational data only. However, this is not central to our claims in the paper and could be left for future work.
>
> ## **Concrete commitments for revision:**
>
> Beyond previous commitments in the original rebuttal, we will make the following specific changes to address the reviewer's concerns:
>
> 1. **Restructure the paper** to lead with the non-fixed topological order results as our primary contribution that demonstrate genuine causal discovery
> 2. **Reframe the fixed-order experiments** as a further analysis that enables detailed mechanistic interpretability (probing and intervening)
> 3. **Expand the Pearl discussion**, given new insights from the rebuttal/discussion period. We will make it clear that the controversy is not about whether statistical data can be used to answer certain causal queries, but rather about whether deep neural networks trained using prediction objectives on passive streams of data can build true causal models and inference engines.
> 4. **Provide detailed analysis** on challenges faced by probing/interventional methods in the non-fixed topological order
>
> These changes should strengthen the paper by emphasizing true causal discovery while maintaining the mechanistic insights from our original approach.

---

> > ### Comment · Reviewer_nnXY · 2025-08-06
> >
> > I thank the authors for their informative responses, which have helped clarify further and resolve some issues. I understand the controversy regarding the difference between the authors' vs. Pearl's claims better now and would like to make a final suggestion:
> >
> > If I understand correctly, the main topic of debate is indeed about the type of *learning objective or modelling approach*, not about the nature of the underlying data. That is:
> >
> > > Can causal knowledge (assessed by answering new L2/L3 questions) emerge from a statistical approach like sequence modelling with next-token prediction (as opposed to building a causal world model with an explicit graphical representation), *irrespective of the type of available data*?
> >
> >  If Pearl's claim is indeed that this should be impossible, even when the statistical model is trained on L2 & L3 data as in the present study, then I think the authors have presented a convincing counterexample. However, I think that **referring to the training data as "passive streams of data" can be seen as misleading/confusing** in this context. Indeed, the data is not actively collected by the learning agent, and is treated as a passive stream by the sequence model. However, in terms of the underlying true causal model (the SCM used for observation, or the physical world more generally), the data is actually interventional or counterfactual in nature. Yet, using the term "passive" suggests that only L1 data is used for learning. In other words, **it is important to distinguish between (i) the actual type of data in the context of the underlying causal model and (ii) how this data is used or treated by the learner** (here, as a passive stream of tokens).
> >
> > I believe that emphasising these nuances would help further clarify the exact claims. (I am aware the authors have already attempted this in the original submission and suggested further improvements during the rebuttal period, but perhaps the above take can be helpful nonetheless.)
> >
> > Overall, the authors' responses and commitments for the revision have addressed my main concerns, and I believe that this paper (in its revised form) will be a valuable addition to the literature on the causal capabilities of LLMs. I will raise my score to 4. My only worry is that the changes resulting from the rebuttal phase are rather major and may benefit from another review cycle.

---

> > > ### Author Response · Authors · 2025-08-06
> > >
> > > We thank the reviewer for this final suggestion. Indeed, we do not disagree with Pearl that models need L2/L3 data (at least in implicit form) to learn the true causal model, as per Pearl's Causal Hierarchy (PCH). "Passive data" is only a contrast to data collected by an agent that is interacting with the world through interventions; "statistical prediction objective" is meant as a contrast to modeling approaches that explicitly define all levels of the hierarchy and explicitly define the operations on those levels. Both the "passive" nature of data collection as well as the "statistical" nature of the algorithm seem to be component's of Pearl's argument. The distinction between type of data used and treatment of that data by the learner is very useful. We will integrate it in the revision.
> > >
> > > We understand the worry that changes seem major. However, we think the _main_ changes should be relatively straightforward: (1) given new insights from the review period, we will add nuances regarding our position/motivation in the introduction/discussion; (2) we will incorporate results without a fixed topological order.
> > >
> > > Finally, we are truly grateful for the high quality review of our work and all the useful feedback during the discussion.

---

### Note · Authors · 2025-08-13

We are grateful to all reviewers for their constructive feedback that has strengthened our work.

## Key strengths recognized

Reviewers consistently appreciated our **experimental design and comprehensive evaluation approach** (Y1qU, d6ju, nnXY). They noted that our work goes beyond behavioral testing to include **mechanistic interpretability** (Y1qU, d6ju, m4HY). The **novel integration of causality with next-token prediction** was recognized as addressing fundamental theoretical questions about Pearl's claims (Y1qU, nnXY, d6ju). Finally, they thought the paper was **well written/clear** (nnXY, Y1qU, m4HY, d6ju).

## Fixed topological order addressed

The most significant criticism came from Reviewer nnXY, who correctly identified that our original results used a **fixed topological variable order** ($V_1 \prec V_2 \prec V_3 \prec V_4$). We addressed this by **successfully retraining models without fixed topological order**, demonstrating that our approach works for genuine causal discovery where both structure and parameters must be learned per SCM. **Reviewer nnXY acknowledged this addition as "very valuable and important"** and raised their score accordingly.

## Clarifications on claims and scope

Several reviewers sought clarity on our **positioning relative to Pearl's theoretical claims** (nnXY, m4HY, uXJf). We provide an existence proof challenging Pearl's impossibility claim that deep neural networks trained using statistical prediction objectives cannot reach models/reasoning at interventional (L2) and counterfactual (L3) levels.

Regarding our **constrained experimental setting**, reviewers noted limitations of linear Gaussian SCMs and artificial language (d6ju, Y1qU). However, this setup enabled rigorous evaluation with ground truth validation. As an existence proof against a universal theoretical claim, one well-controlled counterexample provides valuable evidence.

## Revision commitments

Post-rebuttal, **three reviewers raised their scores** (nnXY, d6ju, uXJf), while **reviewer m4HY maintained their positive assessment**. We will:

- **lead with non-fixed topological order** results as our primary contribution
- **expand our discussion** of Pearl's claims and add appropriate cautions against over-interpreting results for real-world LLM capabilities

This work provides evidence that true causal models and inference engines can emerge through statistical prediction—a foundational contribution to understanding neural network capabilities.

---

### Decision · Program_Chairs · 2025-09-17

**Decision:**

Accept (poster)

**Comment:**

This paper tackles a foundational question, whether statistical learning systems can achieve interventional (L2) and counterfactual (L3) reasoning.
Pearl says no.
The authors propose a simplified setting, where a large number of 4-variable structural causal models (SCMs) is used to generate L2 sentences and L3 sentences. For some SCMs, L2 and L3 sentences are used in the train; for other (referred as test thereafter), only L2 sentences are used in the train.
The proof of concept is established by showing that the trained LLM successfully answers L3 queries on test SCMs.
In brief, the authors say yes.

The value of this paper thus lies - besides tackling a hot question - in proposing an innovative artificial methodology, that imo could be applied to investigate other limitations of LLMs, as this methodology avoids the known pitfall that LLMs can find the sought answer in the data they have access to.

The discussion about the merits and value of the paper has been long and thorny.
Rev nnXY spotted an issue in the experimental setting, concerning the fixed causal ordering of the variables. This issue was fixed by the authors.
Rev Y1qU is wondering whether there might be a leakage between the train and the test, given the restricted experimental setting.
Rev m4HY, though positive, states that related work should have been mentioned and discussed (with a list of references).
Rev d6ju is concerned by the highly constrained and simplified experimental setting. Authors reply that they only aim to refute Pearl's claim; and a negative example (here, a proof of concept for successful L3 reasoning) is enough. While d6ju globally agrees, Reviewer insists on the fact that the discussion about the limitations of this proof of concept (e.g., when natural language is considered instead of artificial one) be detailed and careful.
Rev uXJf is concerned with the exact contribution of the paper, as the state of the art already shows that statistical data may be enough to answer causal questions or identify the causal model. Authors state that their goal and their contribution is providing an empirical counterexample to Pearl’s impossibility claim.